# BINARY NEURAL NETWORK FOR HYPERSPECTRAL PAN-SHARPENING

## ABSTRACT

Hyperspectral pan-sharpening aims to generate high-resolution hyperspectral (HRHS) images by fusing low-resolution hyperspectral (LRHS) data with high-resolution panchromatic (PAN) images, enabling applications in mapping, surveillance, and environmental monitoring. While deep learning methods achieve strong performance, their heavy computational and memory demands limit deployment on resource-constrained satellite platforms. To address this, we explore binary neural networks (BNNs) for hyperspectral pan-sharpening. Conventional binarization, however, introduces gradient instability and severe information loss, compromising spectral spatial fidelity. We propose the Adaptive Tan Identity Straight-Through Estimator (ATISTE), a soft binarization strategy that decouples forward approximation from gradient propagation and employs adaptive scaling to preserve consistency with full-precision features. Building on ATISTE, we design HS-BiNet, a lightweight binary CNN with residual connections and multi-scale fusion, to effectively capture spectral–spatial dependencies while avoiding computationally intensive operations such as unfolding inference and non-local self-attention. This ensures suitability for real-time deployment on edge and satellite platforms. Extensive experiments show that HS-BiNet consistently outperforms binary baselines and remains competitive with, and in some cases surpasses, full-precision models, offering a practical solution for high-fidelity HRHS reconstruction.

## 1 INTRODUCTION

High-resolution hyperspectral (HRHS) images are vital in numerous remote sensing applications, including mapping services, military surveillance, and environmental monitoring Ram et al. (2024); Sharma et al. (2020); Pallas Enguita et al. (2024); Carvalho et al. (2019); Stuart et al. (2019); Bedini (2017), due to their superior spectral fidelity and fine spatial detail. However, directly acquiring HRHS images remains a technical challenge, as existing remote sensing sensors are limited in their ability to capture high spectral and spatial resolutions simultaneously. To mitigate this constraint, hyperspectral pan-sharpening has emerged as a reliable solution, whereby low-resolution hyperspectral (LRHS) images are fused with high-resolution panchromatic (PAN) images to generate enhanced HRHS outputs. This fusion process exploits the rich spectral information of HS data and the spatial granularity of PAN images, producing outputs that approximate true HRHS quality.

In recent years, deep learning (DL)-based methods have become the dominant approach for hyperspectral pan-sharpening, surpassing traditional techniques Cai & Huang (2020); Yang et al. (2017). Early models like HyperPNN He et al. (2019) introduced CNNs specifically designed for hyperspectral data, with spectral encoders, decoders, and spatial-spectral fusion modules. Building on this, HSpeNet Hu et al. (2022) added preprocessing subnets for enhanced PAN feature extraction, DenseNet-inspired fusion, and spectral-aware loss functions to balance spatial detail and spectral consistency. Further advances include DHP-DARN Zheng et al. (2020) and DIP-HyperKite Bandara et al. (2021), which use deep image prior (DIP) methods to regularize spectral upsampling and improve fusion under limited data. Notably, DIP-HyperKite's inverse U-Net enables spatial over-expansion, enhancing high-frequency detail beyond PAN resolution. Most recently, HyperD-SNet Zhuo et al. (2022) combined handcrafted edge detectors (e.g., Sobel, Prewitt) with multiscale deep-shallow feature extraction and per-band Spectral Attention for precise, adaptive detail injection across hyperspectral channels. Since then, a flood of increasingly complex network architectures has

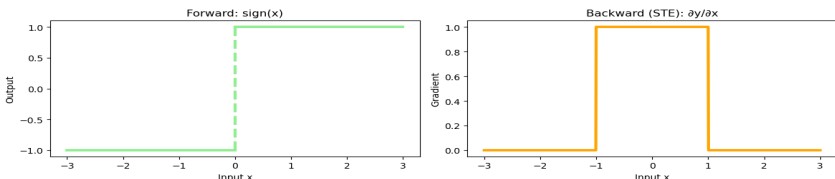

Figure 1: Illustration of Forward and Backward Propagation for the Sign Function.

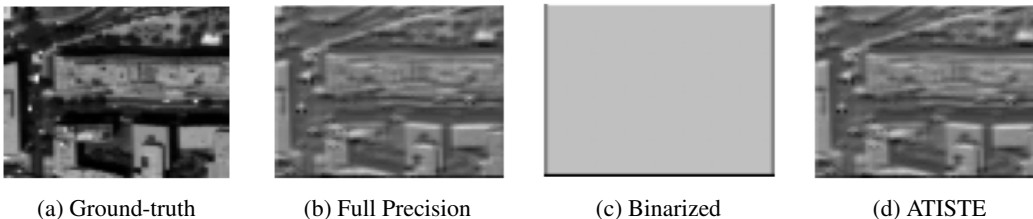

| (a) Ground-truth | (b) Full Precision | (c) Binarized | (d) ATISTE |

Figure 2: Analysis of binarization: Given the ground-truth (a) and full-precision feature map (b), the corresponding binarized activation (c) exhibits significant degradation, while ATISTE outputs (d) retain more information and better preserve the original features.

been proposed to improve further pan-sharpening outcomes, including multi-scale designs, adaptive frameworks, spatial-frequency representations, and transformer-based models, all contributing to richer spatial-spectral fusion capabilities Bandara & Patel (2022); Dong et al. (2021); Sun et al. (2024); Guan & Lam (2021); Wu et al. (2022); Feng et al. (2024). Despite their strong performance, these state-of-the-art models often demand considerable computational resources, such as high-end GPUs, making them impractical for deployment on resource-constrained satellite platforms. Driven by this challenge, we explore network binarization for hyperspectral pan-sharpening to reduce computational complexity while substantially maintaining high reconstruction fidelity. We begin by analyzing the transformation of real-valued activations under the binarization process. As illustrated in Fig. 2, binarized feature representations suffer from significant information loss compared to their full-precision counterparts. This degradation is especially critical in hyperspectral pan-sharpening, where pixel-level accuracy is essential to preserve spatial detail and spectral integrity. Conventional binarization approaches often fail to maintain this fidelity, leading to suboptimal reconstruction quality. Despite significant progress, prior works on quantization/binarization and surrogate gradient methods for BNNs reveal consistent limitations. Early binarization approaches such as XNOR-Net Rastegari et al. (2016b) enabled efficient binary convolutions but suffered large accuracy drops due to crude sign-based binarization and gradient mismatch. Follow-up studies identified forward–backwards inconsistency, where the forward sign function is discontinuous (as shown in Fig. 1), while backwards estimators (e.g., identity mapping in classic STE) introduce severe bias and vanishing gradients Yin et al. (2019). Attempts to close the accuracy gap in low-bit and binary networks, such as 2-bit quantization methods Choi et al. (2018) and differentiable soft quantization Gong et al. (2019b), improved forward approximation but often relied on rigid functional forms or hand-designed smooth surrogates that could not dynamically adapt, leading to unstable gradients and limited scalability. More advanced strategies, such as information retention Qin et al. (2020c), coupled binary activations Kim et al. (2020), and reviving dead weights Xu et al. (2021b), addressed specific failure cases such as gradient starvation or inactive parameters, but typically introduced additional heuristics or complex mechanisms without fully resolving the fundamental trade-off between gradient stability and binarization fidelity. Even the recent Wu et al. (2023), which reframes STE through an equilibrium perspective, still couples approximation accuracy and gradient stability via a single parameter, causing instability when scaled to deeper architectures or larger datasets. Collectively, these works expose four persistent issues: forward–backwards mismatch, gradient vanishing or explosion, rigid or over-parameterized surrogates with poor adaptability, and unstable convergence in deeper networks.

To overcome these limitations, we introduce Adaptive Tan Identity Straight-Through Estimator (ATISTE), a soft binarization strategy based on a novel straight-through estimator that balances binarization fidelity with gradient stability. Our STE employs a dual-path surrogate formulation: one path leverages a smooth, saturating function (a bounded tanh variant) to stabilize gradients during backpropagation (mitigating gradient vanishing or explosion), and the second path preserves a

sharp approximation for practical forward inference (reducing forward–backwards mismatch). This decoupling enables independent control over the approximation in each pass, ensuring effective gradient flow even in saturated regions and improving convergence stability in deeper networks. Furthermore, STE incorporates an adaptive channel-wise scaling mechanism derived from activation statistics, aligning the magnitude of binary activations with their real-valued equivalents (addressing rigidity of fixed surrogates and reducing information loss). This mechanism further enhances adaptability, making STE particularly suitable for high-precision tasks like hyperspectral fusion.

Building on STE, we develop HS-BiNet, a compact and efficient convolutional architecture specifically tailored for hyperspectral pan-sharpening. HS-BiNet employs a modular structure in which key convolutional layers are binarized using STE-enhanced binary convolutions. We incorporate residual connections and intermediate skip links to strengthen information propagation and preserve contextual awareness. Additionally, the network leverages lightweight local fusion units and multi-scale receptive fields to effectively capture the intricate spectral and spatial dependencies characteristic of hyperspectral imagery. This combination of STE-driven binarization and architectural refinements enables HS-BiNet to maintain high reconstruction fidelity while significantly reducing bitwidth and computation. Extensive experiments confirm that HS-BiNet consistently surpasses conventional binary baselines across all key evaluation metrics and narrows the performance gap with full-precision models. In several cases, it even outperforms full-precision methods, demonstrating the practical viability and strength of the proposed approach for real-world hyperspectral fusion tasks.

The main contributions of this work are summarized as follows:

1. We propose a novel straight-through estimator that introduces a dual-path gradient approximation strategy and an adaptive scaling mechanism, significantly improving the stability and accuracy of binarized neural networks. This work presents the first known application of binary neural networks to hyperspectral pan-sharpening.

2. We design HS-BiNet, a lightweight binary CNN architecture tailored for hyperspectral pan-sharpening, which integrates ATISTE-based binary convolutions with residual and multi-scale modules to ensure effective spectral-spatial feature learning.

3. We demonstrate that our binarized network achieves state-of-the-art performance among binary models and remains competitive with, and even surpasses, many full-precision models, all while drastically reducing memory and computation.

## 2 RELATED WORK

### 2.1 DEEP LEARNING-BASED PAN-SHARPENING

In recent years, deep learning has made remarkable advances in low-level vision tasks, profoundly impacting pan-sharpening, where DL methods now lead He et al. (2023; 2024). HyperPNN He et al. (2019) pioneered the use of compact CNNs for hyperspectral fusion, competing well with traditional optimization-based methods. Hyper-DSNet Zhuo et al. (2022) introduced band-wise processing to handle inter-band variability, while DIP-HyperKite Bandara & Patel (2022) combined Deep Image Prior upsampling with an over-complete HyperKite network to learn residual high-frequency details under spatial and spectral constraints. Subsequently, the field rapidly advanced with architectures like AIDB-Net Sun et al. (2024), employing adaptive information distillation blocks to enhance key spectral and spatial features, and FPFNet Dong et al. (2023), which used progressive feature fusion for improved detail recovery. LPPNet Dong et al. (2021) further improved effectiveness by integrating local and global priors. Moving beyond CNNs, transformer-based models, such as HyperTransformer Bandara & Patel (2022), apply self-attention to capture long-range spectral dependencies, while diffusion models target enhanced fusion quality. Despite their strong performance, these methods incur high computational and memory costs, which restricts their deployment on resource-limited satellite platforms. To tackle this, our work investigates binarized neural networks as a lightweight and effective alternative for pan-sharpening.

### 2.2 BINARY NEURAL NETWORK

Binarization is the most extreme form of model quantization, compressing networks by restricting both weights and activations to binary values of $-1$ and $+1$, which significantly reduces storage

consumption and computational cost. Hubara et al. Hubara et al. (2016) introduce the first binarized neural network by directly quantizing parameters during training. Building upon this idea, Rastegari et al. Rastegari et al. (2016b) incorporate learnable scaling factors in XNOR-Net, achieving up to a $58\times$ acceleration and a $32\times$ reduction in memory requirements. Motivated by these efficiency gains, recent works have increasingly explored binary networks for various low-level vision tasks Cai et al. (2023); Chen et al. (2024); Jiang et al. (2023); Song et al. (2023); Xin et al. (2020); Xia et al. (2022). Xin et al. Xin et al. (2020) develop the first fully binarized super-resolution model by binarizing both weights and activations. Jiang et al. Jiang et al. (2021) further propose a binary training framework that removes batch normalization layers, while Xia et al. Xia et al. (2022) design a basic binary convolution unit for image restoration through detailed component analysis. Frequency decomposition is separately handled in Jiang et al. (2023), and Chen et al. Chen et al. (2024) introduce a binarized diffusion model for image SR. Remote sensing satellites, which operate under strict constraints on computation, memory, and energy, present a natural need for highly compact models, and binarization provides a promising solution for efficient onboard processing. However, applying conventional binary networks directly to hyperspectral imaging remains challenging. Hyperspectral data consist of many narrow and continuous spectral bands, and the strong quantization in BNNs causes severe information loss that is far more harmful for spectral fidelity than in RGB imagery. Their limited representation capacity also makes it difficult to capture the complex spectral, spatial relationships present in hyperspectral scenes Guerri et al. (2024); Tang et al. (2024). Moreover, the reduced dynamic range of binary activations can distort fine spectral variations and degrade the accuracy of reconstructed spectral curves Hou et al. (2025). Training is further complicated by the non-smooth nature of binarization Qin et al. (2020a). Binary constraints can also weaken band-wise spectral consistency, which is critical for downstream tasks such as material classification and anomaly detection. Due to the high dimensionality and strong inter-band correlations in hyperspectral images, binary networks often lack sufficient expressive power to generalize well. As a result, although binary networks offer significant efficiency benefits, their direct application to hyperspectral imaging faces notable limitations that must be addressed for effective hyperspectral fusion.

## 3 METHODOLOGY

### 3.1 OVERALL ARCHITECTURE

As illustrated in Fig. 3, the overall framework of our proposed *HS-BiNet* consists of four main components: (1) an Multi-Scale Feature Extractor (MSFE), (2) an Edge Injector Module, (3) a Fusion Module, and (4) a Decoder Module. All components are built on binary convolution (detailed in APPENDIX A.1), implemented with our Straight-Through Estimator (STE) to enable efficient training and inference while preserving representational capacity. Given an input low-resolution hyperspectral image $\mathbf{HS} \in \mathbb{R}^{H \times W \times B}$ with $B$ spectral bands and a high-resolution panchromatic image $\mathbf{P} \in \mathbb{R}^{sH \times sW}$ with $s$ as the spatial upsampling factor, we first apply a deformable convolution-based Dai et al. (2017) upsampler to the hyperspectral image to match the spatial resolution of $\mathbf{P}$, producing the initial feature map $\mathbf{F}_0 \in \mathbb{R}^{sH \times sW \times B}$. The upsampled hyperspectral input is then processed by the MSFE module for spectral feature extraction, where multi-scale features are obtained using binary convolutions with kernel sizes $3 \times 3$, $5 \times 5$, and $7 \times 7$. The resulting feature maps are concatenated along the channel dimension to form a multi-scale feature representation $\mathbf{F}_{ms} \in \mathbb{R}^C$, where $C$ represents the hidden states. A subsequent $1 \times 1$ binary convolution reduces the concatenated features from $3C$ to $C$ channels, followed by a residual connection from the input to preserve the original spectral content during multi-scale fusion. This design enriches the representation with multi-scale contextual information while maintaining constant channel size and low computational cost.

Following the multi-scale extraction, the MSFE employs Binary Residual Blocks, each consisting of two binarized $3 \times 3$ convolutions with stride 1 and padding 1, each followed by batch normalization. A residual connection from the block input to its output preserves spectral information across layers and mitigates degradation. This residual formulation enables the block to learn corrective transformations rather than complete mappings, improving training stability and reconstruction accuracy. At each stage of processing, the Edge Injector applies a binary convolution to the PAN image, producing spatial features $\mathbf{F}_{sp} \in \mathbb{R}^C$. This transforms the single-channel PAN input into a multi-channel feature representation that highlights spatial details for integration with the spectral features extracted by the MSFE. The Fusion unit receives $\mathbf{F}_{sp}$ from the edge injector and $\mathbf{F}_{spc}$ from

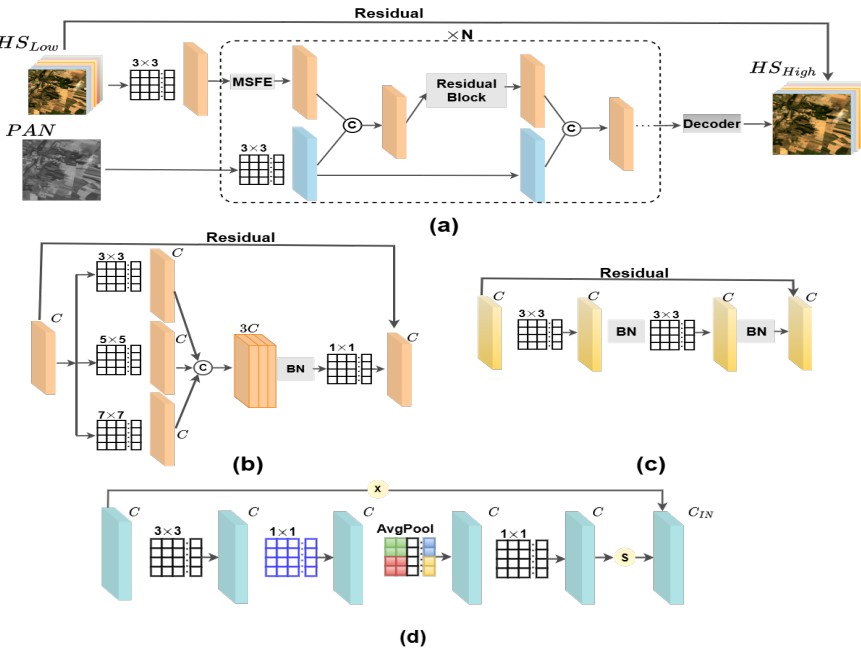

Figure 3: The framework of the proposed HS-BiNet, with hidden dimension $C$, comprises four key components: (a) overall network architecture, (b) multi-scale feature extractor, (c) residual block, and (d) decoder block, where ©  denotes concatenation of feature maps and BN refers to BatchNorm. In the illustration, black-colored kernels represent the binary convolution layers, while blue-colored kernels indicate the full-precision convolution layers.

the MSFE, concatenating them along the channel dimension to form a combined spectral-spatial representation. This representation is then processed by a binary convolution to project it back to the target feature dimension $C$, ensuring compatibility with subsequent stages while preserving spectral fidelity and spatial sharpness.

Finally, the Decoder refines the fused spectral-spatial representation to produce the high-resolution hyperspectral output $\hat{\textbf{HS}}$. The input feature map $\textbf{F}_{in} \in \mathbb{R}^C$ is first processed by a binarized $3 \times 3$ convolution to enhance both spatial and channel-level details, followed by a $1 \times 1$ convolution to adjust channel dimensionality and reduce computational complexity. A lightweight enhancement mechanism then emphasizes the most informative features while suppressing less relevant responses. This mechanism begins with adaptive average pooling to produce a compact descriptor for each channel, followed by a binarized $1 \times 1$ convolution and a sigmoid activation to compute scaling factors in the range $[0, 1]$. These factors are multiplied element-wise with the original feature map to produce the enhanced output $\textbf{F}_{out}$. A global residual connection adds the upsampled hyperspectral input $\textbf{HS}$ to the decoder output, resulting in the final high-resolution hyperspectral image $\hat{\textbf{HS}}$ that preserves the original spectral signatures while effectively integrating the spatial information from the panchromatic image.

## 3.2 STRAIGHT THROUGH ESTIMATOR

Binarized neural networks face a fundamental challenge in balancing forward fidelity with gradient stability. Naïve binarization Qin et al. (2020a) through the discontinuous sign function introduces severe forward–backward mismatch, where the forward mapping discards critical information while identity-based backward surrogates bias gradients and collapse outside a narrow band. This mismatch is further compounded by gradient starvation: hard or overly sharp surrogates yielding vanishing derivatives in saturation regions or unstable spikes near zero, both of which hinder optimization in deeper architectures. Attempts to address these limitations through smooth approximations such as HardTanh, SoftSign, or Log-Tanh reduce discontinuity but either remain too soft to enforce effective binarization or too rigid to maintain stable gradients. Methods such as ReSTE Wu et al. (2023) struggle when applied to high-dimensional or multi-channel data, where different channels often re-

quire different levels of gradient stability and approximation accuracy. In hyperspectral fusion, this challenge becomes much more severe because the task demands extremely fine spectral precision across hundreds of closely spaced bands. The adaptive parameter in ReSTE, which is intended to balance approximation fidelity and gradient stability, cannot satisfy these conflicting requirements across all spectral channels, leading to suboptimal approximations and unstable gradient flow. Even small residual estimator bias or remaining gradient noise can distort subtle spectral curves, and the progressive schedule of ReSTE, while improving stability, still fails to preserve the delicate spectral details required for accurate hyperspectral reconstruction. As a result, although ReSTE is more stable than standard STE methods, it remains insufficient for maintaining high spectral fidelity and continuous-value accuracy in hyperspectral imaging. The proposed ATISTE addresses the fundamental trade-off between forward approximation fidelity and gradient stability that limits existing surrogate estimators in BNN training. Its formulation integrates a smooth, saturating nonlinearity with a linear residual, resulting in a flexible surrogate that guarantees non-zero gradients and bounded variance. Formally, for an input activation $x \in \mathbb{R}$, the forward surrogate is defined as:

$$f_{\alpha,\lambda}(x) = (1-\lambda)\tanh(\alpha x) + \lambda x, \tag{1}$$

where the sharpness parameter $\alpha > 0$ controls the steepness of the nonlinear path, and the residual weight $\lambda \in [0,1]$ adjusts the contribution of the identity path. This design enables smooth interpolation between two established estimation strategies. Specifically, setting $\lambda = 0$ reduces the estimator to a pure $\tanh(\alpha x)$, acting as a smooth approximation of the sign function, while $\lambda = 1$ recovers the identity mapping, corresponding to the classic STE backward surrogate. By selecting intermediate $\lambda$ values, ATISTE seamlessly interpolates between these extremes, allowing a controlled transition from soft to hard binarization during training. The backward pass supplies the pseudo-gradient used in optimization, derived as:

$$\frac{\partial f_{\alpha,\lambda}(x)}{\partial x} = (1-\lambda)\alpha\left(1 - \tanh^2(\alpha x)\right) + \lambda. \tag{2}$$

Equation 2 reveals two critical structural terms. The first term, the scaled derivative of $\tanh$, is sharply localized near zero and facilitates learning in the binarization-critical region. The second term, a constant gradient floor of magnitude $\lambda$, ensures non-zero gradients even when $|x|$ is large and the $\tanh$ path saturates. Unlike hard sign surrogates, which suffer from vanishing gradients outside $[-1,1]$, ATISTE maintains informative gradients everywhere. Furthermore, the boundedness of the derivative is expressed as:

$$\lambda \leq f'_{\alpha,\lambda}(x) \leq (1-\lambda)\alpha + \lambda. \tag{3}$$

This dual bound prevents gradient starvation (via the non-zero floor $\lambda$) and gradient explosion (via the finite ceiling $(1-\lambda)\alpha + \lambda$). Such properties offer a clear advantage over power-law surrogates, whose derivatives can either diverge near zero or vanish in saturation regions. From the forward perspective, ATISTE is rational: its approximation error is no worse than that of the classic STE. Defining the mean-squared error relative to the ideal sign function as:

$$E(\alpha,\lambda) = \mathbb{E}\left[\left(\operatorname{sign}(x) - f_{\alpha,\lambda}(x)\right)^2\right], \tag{4}$$

we can analyze its asymptotic behavior. As $\alpha \to \infty$, the nonlinear component converges pointwise to the sign function:

$$\lim_{\alpha \to \infty} f_{\alpha,\lambda}(x) = (1-\lambda)\operatorname{sign}(x) + \lambda x, \tag{5}$$

yielding:

$$\lim_{\alpha \to \infty} E(\alpha,\lambda) = \lambda^2 \mathbb{E}\left[\left(\operatorname{sign}(x) - x\right)^2\right] \leq E_{\text{STE}}. \tag{6}$$

Thus, for all $\lambda < 1$, ATISTE achieves strictly lower or equal approximation error compared to the identity-based STE.

A key feature of ATISTE lies in its tunable bias–variance trade-off. Increasing $\alpha$ sharpens the forward mapping, reducing bias by making $f_{\alpha,\lambda}$ more sign-like, while increasing local gradient variance. Conversely, increasing $\lambda$ strengthens the residual path, raising the gradient floor and reducing variance, but at the cost of forward bias. The decoupling into two independent parameters provides explicit control over optimization dynamics, enabling schedules where $\alpha$ is gradually increased and $\lambda$ decreased over training epochs. This ensures smooth convergence to a strongly binarized model without starving gradients. Because of its dual-path design, ATISTE naturally integrates bounded non-zero gradients (Eq. 3) with a rational forward approximation (Eqs. 4, 5, 6) and

Table 1: Quantitative comparison of our HS-BiNet with representative full-precision and binary methods on the Botswana and WDC datasets. Here, **US** denotes unsupervised methods and **S** denotes supervised methods.

| Category | Method | Type | Botswana | | | | | Washington DC Mall (WDC) | | | | |
|---|---|---|---|---|---|---|---|---|---|---|---|---|
| | | | PSNR | CC | SSIM | SAM | ERGAS | PSNR | CC | SSIM | SAM | ERGAS |
| Full-Precision | GLP | US | 32.559 | 0.951 | 0.837 | 1.383 | 1.207 | 27.946 | 0.934 | 0.761 | 6.546 | 5.110 |
| | GSA | US | 31.739 | 0.939 | 0.828 | 1.389 | 1.386 | 24.462 | 0.906 | 0.671 | 7.846 | 6.079 |
| | CNMF | US | 30.220 | 0.917 | 0.788 | 1.934 | 1.718 | 24.604 | 0.890 | 0.678 | 8.441 | 6.682 |
| | HySure | US | 30.610 | 0.928 | 0.796 | 1.747 | 1.595 | 25.598 | 0.913 | 0.718 | 7.254 | 5.834 |
| | HyperPNN | S | 33.114 | 0.961 | 0.873 | 1.366 | 1.195 | 29.258 | 0.945 | 0.860 | 4.051 | 5.749 |
| | HSpeNet1 | S | 31.746 | 0.942 | 0.844 | 1.456 | 1.663 | 29.634 | 0.960 | 0.870 | 4.039 | 4.266 |
| | HSpeNet2 | S | 32.575 | 0.953 | 0.849 | 1.400 | 1.348 | 29.700 | 0.961 | 0.872 | 4.009 | 4.261 |
| | FusionNet | S | 32.506 | 0.952 | 0.850 | 1.397 | 1.367 | 29.696 | 0.959 | 0.866 | 3.917 | 4.339 |
| | Hyper-DSNet | S | 33.538 | 0.964 | 0.876 | 1.305 | 1.126 | 30.232 | 0.964 | 0.875 | 4.102 | 3.943 |
| | FPFNet | S | 33.451 | 0.962 | 0.871 | 1.369 | 1.135 | 30.291 | 0.957 | 0.855 | 4.440 | 4.250 |
| | FusionMamba | S | 33.943 | 0.966 | 0.881 | 1.277 | 1.076 | 31.860 | 0.965 | 0.881 | 3.755 | 3.882 |
| | DM-ZS | US | 39.280 | 0.922 | 0.901 | 1.384 | 1.372 | 38.160 | 0.955 | 0.899 | 4.544 | 6.562 |
| Binary | BNN | S | 23.234 | 0.119 | 0.845 | 3.140 | 5.044 | 13.054 | 0.012 | 0.665 | 18.057 | 17.085 |
| | ReActNet | S | 27.532 | 0.560 | 0.862 | 2.740 | 3.860 | 20.256 | 0.612 | 0.734 | 11.110 | 12.027 |
| | BTM | S | 27.485 | 0.596 | 0.881 | 2.946 | 2.570 | 17.311 | 0.072 | 0.646 | 18.111 | 11.674 |
| | FABNet | S | 23.853 | 0.175 | 0.832 | 3.421 | 3.463 | 14.700 | 0.040 | 0.646 | 18.312 | 12.299 |
| | ReSTE | S | 27.959 | 0.659 | 0.887 | 2.792 | 2.265 | 24.208 | 0.759 | 0.823 | 8.195 | 6.665 |
| | Bi-DiffSR | S | 30.821 | 0.798 | 0.883 | 2.373 | 2.666 | 25.989 | 0.845 | 0.882 | 8.007 | 8.447 |
| | **HS-BiNet** | **S** | **37.285** | **0.900** | **0.891** | **1.622** | **1.852** | **36.003** | **0.947** | **0.885** | **5.468** | **4.781** |
| Ideal values | | | $+\infty$ | 1 | 1 | 0 | 0 | $+\infty$ | 1 | 1 | 0 | 0 |

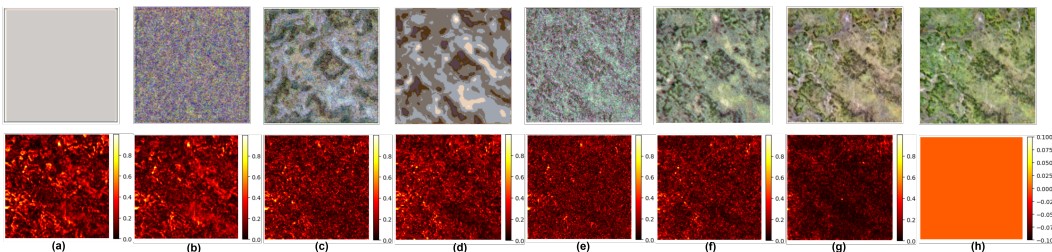

Figure 4: Qualitative comparison of our model with representative binary methods on the reduced-resolution Botswana dataset. (a) BNN, (b) FabNet, (c) ReactNet, (d) ReSTE, (e) BTM, (f) Bi-Diffsr, (g) HS-BiNet, and (h) Ground-truth.

explicit control over the error–stability trade-off. These properties collectively explain the empirically observed faster convergence and higher accuracy of ATISTE in deep BNNs compared to prior single-parameter surrogates. A detailed discussion of ATISTE's rationality, flexibility, estimation error, and gradient instability is provided in Appendix A.2.2.

# 4 EXPERIMENT

In this section, we present the evaluation on reduced-resolution and full-resolution hyperspectral pansharpening. We further provide extensive ablation studies to validate the effectiveness of our STE. Due to page limitations, detailed descriptions of the datasets, baselines, and experimental settings are included in the Appendix A.2.

## 4.1 COMPARISON WITH STATE-OF-THE-ART

We first evaluate the similarity between the fused hyperspectral images and the ground-truth data using reduced-resolution experiments on the Washington DC Mall (WDC) and Botswana datasets. As shown in Table 1, our method clearly outperforms existing binary approaches (BNN Hubara et al. (2016), ReActNet Liu et al. (2020), BTM Jiang et al. (2021), FABNet Jiang et al. (2023), ReSTE Wu et al. (2023), Bi-Diffsr Chen et al. (2024)) across all evaluation metrics. Our model also achieves comparable results to many full-precision baselines (GLP Aiazzi et al. (2006), GSA Aiazzi et al. (2007), CNMF Yokoya et al. (2011), HySure Simoes et al. (2014), HyperPNN He et al. (2019), HSpeNet1/2 Hu et al. (2022), FusionNet Deng et al. (2020), Hyper-DSNet Zhuo et al. (2022), FPFNet Dong et al. (2023), FusionMamba Peng et al. (2024), DM-ZS Xiao et al. (2025)), show-

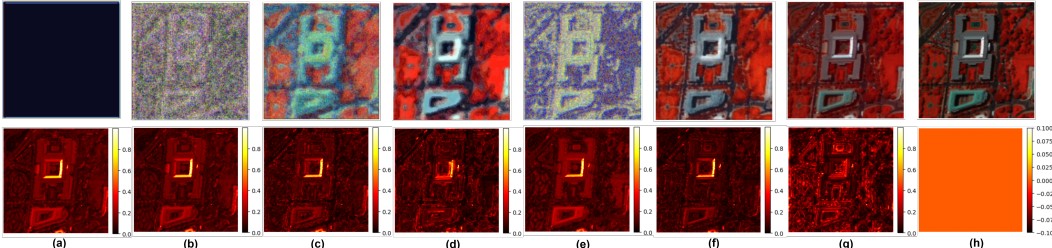

Figure 5: Qualitative comparison of our model against other binary methods on reduced-resolution WDC dataset. (a) BNN, (b) FabNet, (c) ReactNet, (d) ReSTE, (e) BTM, (f) Bi-Diffsr, (g) HS-BiNet, and (h) Ground-truth.

Table 2: Full-resolution evaluation on the FR1 dataset with binary methods.

| Method | $D_\lambda$ | $D_s$ | QNR |
|---|---|---|---|
| BNN | 0.093 | 0.062 | 0.849 |
| ReActNet | 0.318 | 0.009 | 0.675 |
| BTM | 0.117 | 0.041 | 0.846 |
| ReSTE | 0.405 | **0.001** | 0.594 |
| Bi-DiffSR | 0.068 | 0.083 | 0.853 |
| **HS-BiNet** | **0.037** | 0.104 | **0.861** |
| Ideal values | 0 | 0 | 1.00 |

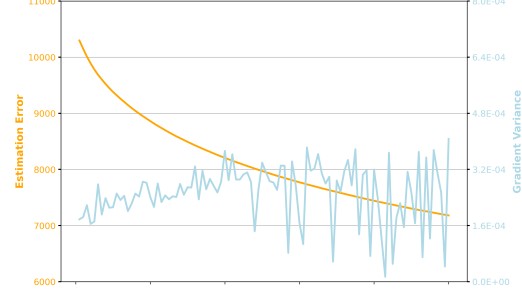

Figure 6: Estimating error (orange curve) and gradient instability (blue curve) indicators on CIFAR-10.

ing strong capability despite binarization. The qualitative results in Figures 4 and 5 further support these findings. RGB visualizations exhibit clearer structures and more accurate colors, while the MAE maps show fewer bright regions, indicating closer agreement with the ground truth. To test generalization in real-world conditions, we also evaluate full-resolution FR1 data from the PRISMA pansharpening contest. As shown in Table 2, our method achieves state-of-the-art results, producing the highest QNR values and competitive distortion measures ($D_\lambda$, $D_s$). It preserves both spectral and spatial details well and provides the efficiency benefits of binarization. Additional analysis of spectral vectors from a WDC sample is presented in APPENDIX A.2.4, where our method shows noticeably lower spectral distortion than other binary networks.

### 4.2 ABLATION: EXPERIMENTS ON STE

To demonstrate the effectiveness of our method, we conduct a comprehensive performance study on the CIFAR-10 dataset, benchmarking against a broad range of state-of-the-art (SOTA) binary networks. Unlike many existing approaches that rely on auxiliary components such as additional modules or loss functions, our method introduces a learnable straight-through estimator that simultaneously adapts both the forward surrogate and its backwards gradient, without requiring any auxiliary modules or losses. For clarity, we mark in Table 3 the methods that depend on such auxiliaries with an asterisk (*). We evaluate three widely adopted backbone architectures, ResNet-18, ResNet-20, and VGG-Small, against diverse SOTA binary methods, including LBA Hou et al. (2016), RAD Ding et al. (2019), DSQ Gong et al. (2019a), Xnor-Net Rastegari et al. (2016a), DoReFa-Net Zhou et al. (2016), IR-Net Qin et al. (2020b), LCR-BNN Shang et al. (2022), RBNN Lin et al. (2020), and FDA Xu et al. (2021a).

As shown in Table 3, our approach consistently achieves the highest Top-1 accuracy across all backbones. For example, with ResNet-18, our method outperforms RBNN* and IR-Net*, both of which rely on auxiliary designs. On ResNet-20, our method surpasses the strongest baseline RBNN*. Similarly, with VGG-Small, our approach outperforms RBNN* and other competing methods. These results highlight the superiority of our method in that, despite eliminating the need for auxiliary modules or loss functions, it consistently outperforms prior SOTA approaches that rely on such components, demonstrating both greater effectiveness and efficiency.

Table 3: Performance comparison with state-of-the-art methods on CIFAR-10. Methods marked with * indicate the use of auxiliary components (either additional modules or loss functions).

| Method | Top-1 Acc. | Method | Top-1 Acc. | Method | Top-1 Acc. |
|---|---|---|---|---|---|
| Full Precision | 94.84 | Full Precision | 91.70 | Full Precision | 93.33 |
| RAD* | 90.50 | DSQ | 84.11 | LBA | 87.70 |
| IR-Net* | 91.50 | DoReFa-Net | 84.44 | Xnor-Net | 89.80 |
| LCR-BNN* | 91.80 | IR-Net* | 85.40 | BNN | 89.90 |
| RBNN* | 92.20 | SLB* | 85.50 | RAD* | 90.00 |
| | | LCR-BNN* | 86.00 | IR-Net* | 90.40 |
| | | FDA* | 86.20 | RBNN* | 91.30 |
| | | RBNN* | 86.50 | | |
| **ATISTE** | **93.00** | **ATISTE** | **87.45** | **ATISTE** | **92.05** |
| ResNet-18 on CIFAR-10 | | ResNet-20 on CIFAR-10 | | VGG-Small on CIFAR-10 | |

Table 4: Benchmark comparison of model complexity and inference performance. All parameter counts are reported in Millions (M).

| Model | Params (M) | FLOPs (G) | Inference (s) |
|---|---|---|---|
| HyperPNN | 0.514 | 7.883 | 0.619 |
| HspeNet | 3.264 | 52.935 | 0.620 |
| DHP-Darn | 9.067 | 12.455 | 1.074 |
| HS-BiNet + ATISTE | 1.60 | 5.599 | 0.565 |
| HS-BiNet + BNN | 1.60 | 5.599 | 0.628 |
| HS-BiNet + Bi-DiffSR | 1.60 | 11.010 | 0.630 |
| HS-BiNet + BTM | 1.60 | 5.599 | 0.585 |
| HS-BiNet + ReActNet | 1.61 | 5.616 | 0.624 |
| HS-BiNet + ReSTE | 1.60 | 5.599 | 0.592 |

### 4.2.1 ANALYSIS ON ESTIMATING ERROR AND GRADIENT STABILITY

Fig. 6 shows a steady decrease in estimation error during training, indicating that the proposed estimator progressively aligns with the sign function and improves forward approximation. At the same time, the gradient variance remains low with only minor fluctuations, demonstrating stable and reliable backward gradients. Together, these trends confirm that the estimator achieves accurate forward approximation while preserving training stability, directly supporting the model's strong performance.

### 4.3 ABLATION: COMPUTATIONAL EFFICIENCY

Table 4 highlights the efficiency of our approach. Across all integration variants, HS-BiNet maintains a lightweight profile with nearly identical parameter counts and FLOPs, while achieving faster or comparable inference times. This demonstrates that HS-BiNet introduces no significant computational overhead. The HS-BiNet + BNN configuration isolates the effect of our architecture within a standard BNN framework, confirming that the observed improvements stem directly from HS-BiNet.

## 5 CONCLUSION

In this paper, we present the first dedicated exploration of binary neural networks for hyperspectral pan-sharpening and introduce HS-BiNet, a lightweight architecture driven by the proposed Adaptive Tan Identity Straight-Through Estimator (ATISTE). Specifically, ATISTE employs a dual-path surrogate formulation that decouples forward approximation from gradient propagation, ensuring stable optimization while reducing information loss. An adaptive channel-wise scaling mechanism is further incorporated to align binary activations with their real-valued counterparts, thereby preserving spectral–spatial fidelity. Building on this estimator, HS-BiNet integrates residual connections, multi-scale receptive fields, and local fusion units to capture the complex dependencies of hyperspectral data within an efficient binary framework. Unlike many recent models, HS-BiNet avoids computationally intensive operations such as unfolding inference and non-local self-attention, making it well-suited for real-time deployment on edge and satellite platforms. With these functional designs, our approach consistently surpasses conventional binary baselines and narrows, or even closes, the gap with full-precision models, demonstrating its practicality for high-fidelity hyperspectral reconstruction on resource-constrained platforms. Upon acceptance, we will release the full source code, trained models, and data-processing scripts to facilitate reproducibility and further research.

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

# A APPENDIX

## A.1 BINARY CONVOLUTION

In model binarization, the surrogate estimator plays a central role in approximating the discontinuous sign function for both weights and activations. However, conventional binary convolutional layers rely on fixed approximations that either over-restrict the representation capacity or destabilize the training process. To address this limitation, we design a customized binary convolutional module, that incorporates channel-wise learnable sharpness parameters and a residual mixing coefficient to flexibly control the trade-off between binarization fidelity and gradient stability.

Specifically, each output channel of the convolutional weights is associated with a learnable sharpness parameter $\alpha_w$, while each input channel of the activations has a corresponding sharpness parameter $\alpha_a$. These parameters determine how aggressively the weights and activations are driven towards their binarized representations. In addition, a residual mixing factor $\lambda$ is introduced and made learnable at the layer level. This factor is constrained within $[0, 1]$ to guarantee stability, and it governs the convex combination of the nonlinear binarization path and the linear passthrough path. In this way, the model adaptively balances strict binary quantization against the retention of full-precision information.

During the forward pass, both weights and activations are binarized using the proposed ATISTE estimator, which approximates the hard sign function in the forward computation while supplying bounded surrogate gradients in the backward pass. By reshaping $\alpha_w$ and $\alpha_a$ for proper broadcasting, the binarization process is applied in a channel-wise manner, ensuring fine-grained control. After binarization, a per-channel scaling factor is computed as the mean absolute value of the real-valued weights. This scaling restores the representational power lost through 1-bit quantization and prevents performance degradation that commonly arises in fully binarized networks.

Finally, the binarized activations and scaled binarized weights are used in a standard convolution operation, inheriting stride, padding, dilation, and grouping from the base full-precision layer. Through this design, the BinaryConv2d module ensures that both weights and activations are quantized with stability-aware surrogates, while maintaining sufficient representational capacity to achieve competitive performance in binarized neural networks.

## A.2 EXPERIMENTAL SETTINGS

To evaluate the effectiveness of our proposed HS-BiNet for remote sensing pansharpening, we conduct extensive experiments on three simulated hyperspectral (HS) datasets[1], i.e., Washington DC, and Botswana, as well as one full-resolution dataset, FR1. The key characteristics of these datasets are summarized in Table 5 for convenient comparison.

1. **Washington DC Mall Dataset**
   This dataset was collected by the Hyperspectral Digital Imagery Collection Experiment (HYDICE) sensor. It originally contains 210 spectral bands covering the range from 0.4 to 2.4 $\mu$m (visible light and near-infrared). After removing unusable bands, 191 bands are retained. The spatial resolution of the data is $1208 \times 307$ pixels. The dataset covers various land cover types, including roofs, streets, gravel roads, grass, trees, water, and shadows. In experimental setups, the Washington DC Mall (WDC) dataset with 191 channels is commonly used. The test set is constructed from four $128 \times 128$ images clipped from the original image, while the remaining portion is utilized for training the network parameters. For training, the original PAN and HS images are partitioned into 921 small patch pairs consisting of $64 \times 64$ PAN patches and $16 \times 16$ HS patches. Additionally, 103 patch pairs are reserved for validation from the simulated patches.

2. **Botswana Dataset**
   Captured by the EO-1 Hyperion sensor over Botswana between 2001 and 2004, this dataset spans a spectral range from 0.4 to 2.5 $\mu$m at 10 nm intervals. Originally comprising 242 spectral bands, 145 bands are preserved after noise removal. The spatial size is $1496 \times 256$ pixels. The dataset includes 14 distinct land cover classes, representing seasonal swamps,

---

[1]Dataset source: `https://github.com/liangjiandeng/HyperPanCollection`

Table 5: Hyperspectral Datasets

| Dataset | Number of Bands | Spectral Range | Spatial Resolution | Image Size | Land Cover Types |
|---|---|---|---|---|---|
| Washington DC | 191 | 0.4–2.4 $\mu$m | 1 m | $1208 \times 307$ | Roofs, Streets |
| Botswana | 145 | 0.4–2.5 $\mu$m | 30 m | $1496 \times 256$ | Seasonal Swamps |
| FR1 | 69 | 0.4–2.5 $\mu$m | 30 m | $2400 \times 2400$ | Roofs, Streets |

occasional swamps, and drier woodlands within the distal portion of the Okavango Delta. In practice, the Botswana dataset is often prepared with 102 effective channels. The test set is composed of four $128 \times 128$ image clips from the original data. For training, the PAN and HS images are divided into 799 overlapping $64 \times 64$ patches, and 168 patch pairs are retained for simulation.

3. **FR1 Full-Resolution Dataset**
   Distributed as part of the PRISMA contest, the FR1 dataset is intended for full spatial resolution pansharpening. The dataset can be downloaded from the official PRISMA website. It contains co-registered PAN and HS images, where a $12\,\text{km} \times 12\,\text{km}$ region (equivalent to $2400 \times 2400$ pixels for the PAN image and $400 \times 400 \times 69$ pixels for the HS image) is extracted from the original $30\,\text{km} \times 30\,\text{km}$ PRISMA acquisition. In experimental settings, the FR1 dataset with 69 channels is employed to evaluate pansharpening performance. The test set consists of two images ($240 \times 240$ for HS and $60 \times 60$ for PAN) clipped from the original image, while the remaining portion is used for training after the downsampling simulation. The training data are divided into 734 small patch pairs, each comprising $60 \times 60$ PAN patches and $10 \times 10$ HS patches, with 82 patch pairs reserved for validation.

Following Wald's protocol, the original HS images from the three simulated datasets serve as the reference (REF) images. Low-resolution hyperspectral (LRHS) images are generated by first applying a Gaussian blur to the reference images, followed by downsampling, where one out of every four pixels is selected in both horizontal and vertical directions. Simulated PAN images are created by applying a suitable spectral response vector to the original reference HS images. The resulting simulated PAN and LRHS images are then used as inputs to various hyperspectral super-resolution methods, including our proposed HS-BiNet. We implemented and trained our model using the Py-Torch framework on an NVIDIA GeForce RTX A5000 GPU with 24 GB of memory. The training employed a batch size of 4, the Adam optimizer, and a cosine annealing warm restarts scheduler. The model was trained for 1,600 epochs on the Washington DC Mall dataset and for 1,000 epochs each on the Botswana and FR1 datasets.

Finally, the super-resolved HS images are quantitatively compared against the original ground-truth HS images using standard quality metrics. To assess the quality of the proposed pansharpening method, we adopt a set of widely accepted spatial and spectral evaluation measures commonly used in hyperspectral image fusion tasks. Specifically, we evaluate performance using PSNR, SSIM, Cross-Correlation (CC), Spectral Angle Mapping (SAM), and ERGAS, as these indices collectively characterize both spatial detail preservation and spectral consistency. The ideal reference values for these metrics are: PSNR $\rightarrow \infty$, SSIM $\rightarrow 1$, CC $\rightarrow 1$, SAM $\rightarrow 0$, and ERGAS $\rightarrow 0$, indicating perfect reconstruction fidelity He et al. (2019); Hu et al. (2022); Deng et al. (2020); Zhuo et al. (2022); Dong et al. (2023). In hyperspectral fusion, such a diverse set of metrics is essential because a method must simultaneously reproduce high-frequency spatial structures and maintain accurate spectral signatures across all bands. PSNR measures overall pixel-wise reconstruction accuracy by evaluating the logarithmic ratio between the signal and the mean squared error; higher values denote lower distortion, although PSNR alone cannot fully capture perceptual or spectral differences. CC complements PSNR by quantifying the linear correlation between spatial patterns of the fused image and the ground truth, thereby reflecting how well spatial structures are retained. SSIM provides a more perceptually grounded spatial assessment by examining luminance, contrast, and structural similarity between corresponding patches, offering insight into the preservation of edges and textures, even though it does not inherently capture spectral distortions. To address spectral fidelity, SAM evaluates the angle between spectral vectors at each pixel, with smaller angles indicating that the fused image preserves the correct spectral shape relative to the ground truth, an essential requirement for tasks such as material classification and unmixing. Finally, ERGAS offers a global, scale-adjusted summary of relative reconstruction error across all spectral bands, integrating both spatial and spectral discrepancies into a single dimensionless score. Together, these complementary

metrics ensure a holistic and reliable evaluation of fusion quality, capturing the aspects of spatial sharpness, structural coherence, and spectral integrity that are fundamental to effective hyperspectral pansharpening.

**Reconstruction Loss.** We adopt the $\ell_1$ loss to supervise the reconstruction of high-resolution hyperspectral images. It is defined as

$$\mathcal{L}_1(x, x_{\mathrm{gt}}) = \|x - x_{\mathrm{gt}}\|_1, \tag{7}$$

where $x_{\mathrm{gt}}$ denotes the ground-truth HR-HSI and $x$ is the reconstructed HR-HSI. We choose the $\ell_1$ formulation because prior studies have shown that it produces sharper and more faithful reconstructions than the $\ell_2$ loss in hyperspectral pansharpening tasks.

The model is optimized using Adam with an initial learning rate of $1 \times 10^{-3}$ and a weight decay of $1 \times 10^{-5}$, while a cosine-annealing warm-restart schedule ($T_0 = 50$, $T_{\mathrm{mult}} = 2$, $\eta_{\mathrm{min}} = 1 \times 10^{-5}$) is employed to periodically refresh the learning rate and stabilize training under binary constraints. No data augmentation is applied, as spatial or spectral transformations can introduce non-physical wavelength distortions that negatively impact hyperspectral fidelity. All binary layers are initialized using Kaiming initialization combined with a scaled real-valued weight formulation, which prevents early saturation during binarization and promotes stable gradient flow in the initial training stages. Furthermore, the entire model is trained from scratch without any pretrained components. Since binary networks are particularly sensitive to learning-rate schedules, initialization strategies, and data integrity, clearly specifying these design choices ensures consistent and fully reproducible training behavior.

### A.2.1 ANALYSIS ON $\alpha$ AND $\lambda$

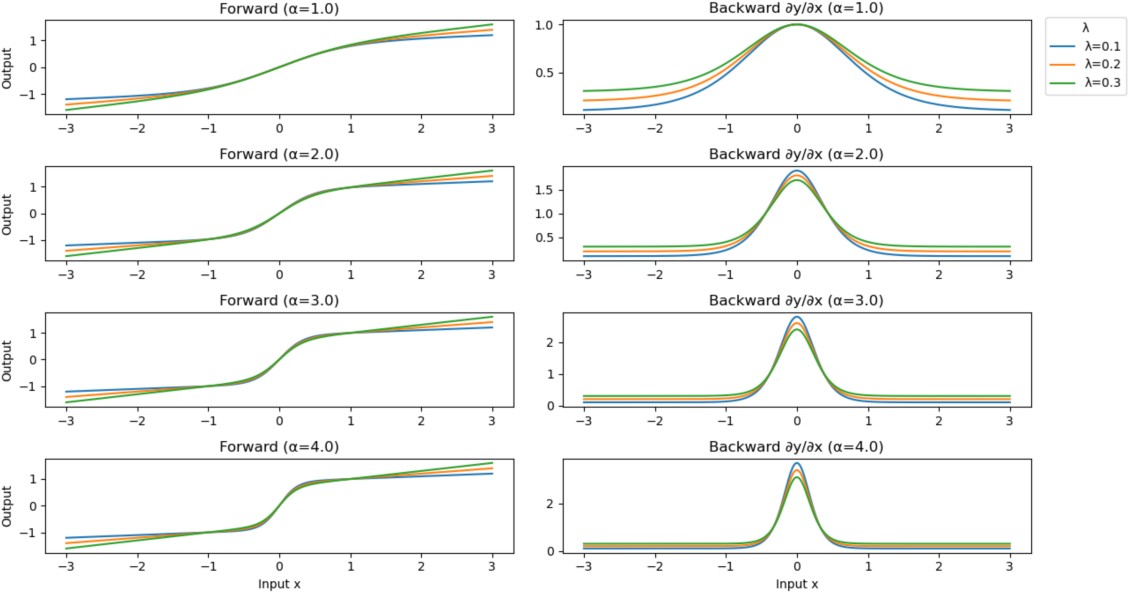

Figure 7: Forward and backward processes with $\alpha = 1.0, 2.0, 3.0, 4.0$ and $\lambda = 0.1, 0.2, 0.3$.

The results presented in Figure 7 illustrate the influence of the scaling factor $\alpha$ and the blending coefficient $\lambda$ on both the forward approximation and the backwards gradient behaviour of the proposed ATISTE. In the forward pass, $\alpha$ regulates the sharpness of the nonlinear surrogate. For smaller values of $\alpha$ (e.g., $\alpha = 1$), the output exhibits a smooth and gradual transition between $-1$ and $+1$, closely resembling a softened activation. As $\alpha$ increases ($\alpha = 4$), this transition becomes significantly steeper, effectively approaching the discontinuous sign function while retaining differentiability. In parallel, $\lambda$ governs the trade-off between the nonlinear surrogate and the identity mapping. Lower values of $\lambda$ emphasize the nonlinear behavior of the tanh-based surrogate. In contrast, higher values introduce a stronger linear component, resulting in forward mappings that deviate less from the input in the saturation regions and preserve more representational continuity.

The backwards pass reveals a complementary dynamic. The derivative with respect to the input, defined as a convex combination of the tanh gradient and the constant identity gradient, demonstrates that $\alpha$ primarily controls the localization of the gradient around zero. Specifically, larger values of $\alpha$ yield sharper, higher-magnitude peaks, enabling stronger learning signals near the decision boundary but diminishing gradient flow elsewhere. In contrast, $\lambda$ lifts the gradient baseline across the domain, preventing complete gradient vanishing in saturated regions and ensuring stable optimization.

The results demonstrate the dual functionality of ATISTE: $\alpha$ governs the degree of binarization sharpness, whereas $\lambda$ regulates the trade-off between expressiveness and trainability. Through the joint effect of these parameters, the estimator achieves a close approximation to binary activations in the forward pass while preserving smooth and reliable gradient propagation in the backwards pass. This property is essential for ensuring convergence and accuracy in binary neural networks.

### A.2.2 PROPERTIES OF ATISTE

**Rationality**   A desirable property of any surrogate estimator is *rationality*, meaning that its forward approximation error should be no worse than the baseline identity-based STE. ATISTE satisfies this property by construction. Recall the forward mapping

$$f_{\alpha,\lambda}(x) = (1 - \lambda)\tanh(\alpha x) + \lambda x, \tag{8}$$

and let the target discrete mapping be $\text{sign}(x)$. The mean-squared forward estimation error can be written as

$$E(\alpha, \lambda) = \mathbb{E}\left[\left(\text{sign}(x) - f_{\alpha,\lambda}(x)\right)^2\right]. \tag{9}$$

As $\alpha \to \infty$, the nonlinear term converges pointwise to the sign function, yielding

$$\lim_{\alpha \to \infty} f_{\alpha,\lambda}(x) = (1 - \lambda)\,\text{sign}(x) + \lambda x. \tag{10}$$

The corresponding asymptotic estimation error becomes

$$\lim_{\alpha \to \infty} E(\alpha, \lambda) = \lambda^2\,\mathbb{E}\left[\left(\text{sign}(x) - x\right)^2\right] \leq E_{\text{STE}}. \tag{11}$$

Thus, for all $\lambda < 1$, ATISTE achieves a strictly lower error than the classic STE, which corresponds to the special case $\lambda = 1$. This demonstrates that ATISTE is rational, as it either matches or outperforms identity-based STE in forward approximation fidelity while maintaining stable gradient flow.

**Flexibility**   In addition to rationality, ATISTE provides *flexibility* through two independent degrees of freedom: the sharpness parameter $\alpha$ and the residual weight $\lambda$. Unlike prior estimators, such as ReSTE, which have a single parameter that simultaneously controls both forward fidelity and gradient dynamics, ATISTE decouples these roles. Specifically, increasing $\alpha$ monotonically sharpens the $\tanh$ component, driving $f_{\alpha,\lambda}(x)$ closer to the discrete sign function:

$$\lim_{\alpha \to \infty} f_{\alpha,\lambda}(x) = (1 - \lambda)\,\text{sign}(x) + \lambda x. \tag{12}$$

Conversely, adjusting $\lambda$ modulates the gradient floor:

$$\lambda \;\leq\; f'_{\alpha,\lambda}(x) \;\leq\; (1-\lambda)\alpha + \lambda, \tag{13}$$

ensuring that gradients remain non-zero everywhere. A higher $\lambda$ provides stronger residual gradients and hence greater stability, while a lower $\lambda$ emphasizes sharper binarization. This independent control over forward fidelity ($\alpha$) and gradient stability ($\lambda$) allows ATISTE to smoothly transition from soft approximations in early epochs to near-hard binarization at convergence, without relying on rigid schedules or heuristic parameter coupling. As a result, ATISTE embodies both rationality, guaranteeing no worse error than STE, and flexibility, enabling adaptive, bounded control of the error–stability trade-off across training.

**Estimation error and Gradient Instability**   A binarized neural network (BNN) relies on surrogate estimators to approximate the discontinuous sign function during training. The effectiveness of a surrogate estimator is governed by two fundamental factors: *estimation error* in the forward pass and *gradient stability* in the backward pass.

The estimation error quantifies the discrepancy between the ideal binarization (the hard sign function) and the chosen surrogate estimator. Formally, for input $z$, an estimator $f(\cdot)$, and distance metric $D(\cdot)$, the error indicator is defined as

$$e = D(\text{sign}(z), f(z)), \tag{14}$$

where we adopt the $L_2$-norm as the distance metric. Thus, the closer $f(z)$ is to $\text{sign}(z)$, the smaller the estimation error.

While low estimation error ensures high forward fidelity, it is equally crucial that the surrogate produces stable gradients in the backward pass. Instability arises when the variance of gradients across parameters is high, which can lead to exploding or vanishing updates. To measure this, we define the gradient instability indicator as

$$s = \text{var}(|g|), \tag{15}$$

where $g$ denotes the parameter gradients, $|g|$ is the element-wise absolute magnitude, and $\text{var}(\cdot)$ denotes the variance operator. This metric captures the divergence in gradient magnitudes, independent of direction.

A critical aspect of ATISTE lies in its treatment of estimation error and gradient stability, the two fundamental axes governing the effectiveness of surrogate estimators in BNNs. From the forward perspective, the surrogate is defined as

$$f_{\alpha,\lambda}(x) = (1 - \lambda)\tanh(\alpha x) + \lambda x, \tag{16}$$

where $\alpha > 0$ controls the sharpness of the nonlinear path and $\lambda \in [0, 1]$ introduces a residual linear contribution. The estimation error with respect to the ideal sign function can be expressed as

$$E(\alpha, \lambda) = \mathbb{E}\left[\left(\text{sign}(x) - f_{\alpha,\lambda}(x)\right)^2\right]. \tag{17}$$

As $\alpha \to \infty$, the nonlinear term approaches the hard sign, yielding

$$\lim_{\alpha \to \infty} f_{\alpha,\lambda}(x) = (1 - \lambda)\text{sign}(x) + \lambda x, \tag{18}$$

and consequently,

$$\lim_{\alpha \to \infty} E(\alpha, \lambda) = \lambda^2 \mathbb{E}\left[\left(\text{sign}(x) - x\right)^2\right] \leq E_{\text{STE}}. \tag{19}$$

This shows that ATISTE is *rational*: its approximation error is guaranteed to be no worse than that of the identity-based STE, and in practice smaller when $\lambda < 1$, thereby ensuring that binarization fidelity improves as training progresses.

From the backward perspective, the pseudo-gradient provided to the optimizer is given by

$$f'_{\alpha,\lambda}(x) = (1 - \lambda)\alpha\left(1 - \tanh^2(\alpha x)\right) + \lambda. \tag{20}$$

Since $0 \leq 1 - \tanh^2(\alpha x) \leq 1$, the pseudo-gradient satisfies the uniform bounds

$$\lambda \leq f'_{\alpha,\lambda}(x) \leq (1 - \lambda)\alpha + \lambda, \tag{21}$$

ensuring both a non-zero gradient floor and a finite ceiling. The lower bound $\lambda$ prevents gradient starvation in saturated regions, while the upper bound $(1 - \lambda)\alpha + \lambda$ controls variance and avoids exploding updates. This boundedness makes ATISTE inherently more stable than classical surrogates such as hard sign, identity, or power-based estimators, which often suffer from vanishing gradients or uncontrolled variance. By decoupling $\alpha$ and $\lambda$, ATISTE provides independent control of binarization fidelity and gradient stability, enabling smooth learning dynamics and effective convergence in deep BNNs.

### A.2.3 ADDITIONAL QUALITATIVE RESULTS

The qualitative results in Figure 8 illustrate that HS-BiNet produces fused hyperspectral images with sharper structures and more faithful spectral appearance in the pseudo-color visualizations, while the absolute error maps show noticeably lower discrepancies from the ground truth compared to other methods.

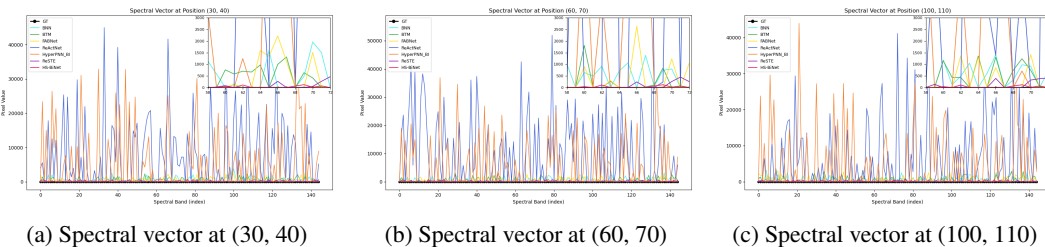

Figure 8: Qualitative evaluation results on the WDC dataset. **Row 1:** Pseudo-color visualizations constructed from spectral bands 20, 50, and 80 of a representative testing sample. **Row 2:** Absolute error maps for spectral band 25 of the same sample, highlighting the difference between the fused result and the ground truth.

Table 6: Component-wise ablation study of HS-BiNet on the test dataset. The impact of removing individual modules on reconstruction performance is reported.

| Model Variant | PSNR ↑ | CC ↑ | SSIM ↑ | SAM ↓ | ERGAS ↓ |
|---|---|---|---|---|---|
| **HS-BiNet** | **37.285** | **0.900** | **0.891** | **1.622** | **1.852** |
| Without Decoder | 36.642 | 0.893 | 0.883 | 1.769 | 1.917 |
| Without MSFE | 36.434 | 0.885 | 0.866 | 1.866 | 1.912 |

### A.2.4 SPECTRAL VECTOR

The spectral vector plots in Figure 9 at positions (30,40), (60,70), and (100,110) indicate that several baseline models produce highly fluctuating and unstable spectral responses. In contrast, the proposed HS-BiNet (red) exhibits smooth, well-scaled reconstructions that remain closely aligned with the ground truth across all spectral bands. Unlike ReActNet and Bi-Diffsr, which produce extreme spikes and unrealistic amplitudes, HS-BiNet avoids such distortions and preserves the natural spectral shape. By effectively zooming, the plots further demonstrate that HS-BiNet captures fine-grained variations within a realistic intensity range, maintaining fidelity to the ground truth without over-amplification. These observations suggest that HS-BiNet achieves more reliable and effective spectral recovery compared to the evaluated methods, effectively balancing reconstruction accuracy and stability.

| (a) Spectral vector at (30, 40) | (b) Spectral vector at (60, 70) | (c) Spectral vector at (100, 110) |
|---|---|---|

Figure 9: Comparison of three spectral vectors extracted from spatial locations (30, 40), (60, 70), and (100, 110) in a WDC testing sample, illustrating the variability of spectral signatures across the scene..

### A.2.5 COMPONENT-WISE ABLATION

The results show that every component of HS-BiNet contributes to reconstruction quality. Removing the Decoder causes a significant drop in PSNR, CC, and SSIM, while increasing SAM and ERGAS, indicating reduced spatial and spectral accuracy. Eliminating the MSFE module results in even greater performance degradation, underscoring its crucial role in effective multi-scale feature extraction. The Edge Injector was not removed in the ablation study because directly adding PAN edge information is ineffective: the PAN image has a single channel, whereas hyperspectral features

have multiple channels, resulting in a dimensional mismatch and poor fusion. The Edge Injector, implemented using binary convolution, is therefore essential for adjusting the channel dimension and enabling proper edge integration. Overall, the complete HS-BiNet achieves the best performance, confirming the necessity and complementarity of all its components.

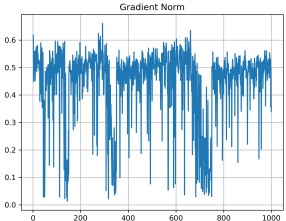
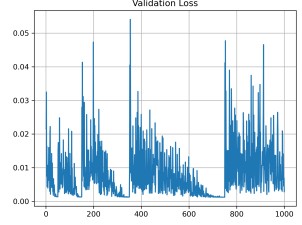
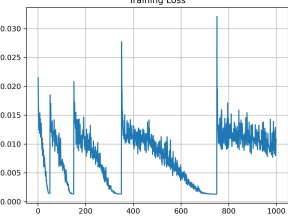

(a) Gradient norm during training.    (b) Validation loss curve.    (c) Training loss curve.

Figure 10: Training dynamics (left-to-right): the gradient-norm evolution indicating stable updates, the validation-loss trajectory showing cyclic patterns from learning-rate restarts, and the training-loss curve demonstrating consistent within-cycle convergence.

### A.2.6    ANALYSIS OF TRAINING DYNAMICS

As seen in Fig. 10b, the validation loss shows small cycles with short spikes at learning-rate changes, but it quickly returns to low values, meaning the model generalizes well. In Fig. 10c, the training loss follows the same pattern, steadily going down within each phase and only jumping when the learning rate resets, showing that the training is stable. The gradient norm in Fig. 10a stays in a safe range without exploding, which confirms that the updates remain stable throughout training. Overall, the three figures demonstrate that the model trains smoothly, handles learning-rate steps correctly, and maintains control over the gradients.

