# OpenReview forum: "Binary Neural Network for Hyperspectral pansharpening"
_ICLR.cc/2026/Conference — Submitted to ICLR 2026_

### Official Review · Reviewer_sXUv · 2025-10-26

**Soundness:** 3
**Presentation:** 1
**Contribution:** 2
**Rating:** 4
**Confidence:** 4

**Summary:**

This paper presents the first exploration of Binary Neural Networks for hyperspectral pan-sharpening, a task that fuses low-resolution hyperspectral images with high-resolution panchromatic images to produce high-resolution hyperspectral outputs. To address the gradient instability and information loss caused by conventional binarization, the authors propose a novel Adaptive Tanh-Identity Straight-Through Estimator . ATISTE decouples forward approximation from gradient propagation via a dual-path design and introduces adaptive channel-wise scaling to maintain fidelity with full-precision features. Building on ATISTE, they design HS-BiNet, a lightweight binary CNN with residual connections and multi-scale fusion modules, avoiding expensive operations like non-local attention or unfolding inference. Extensive experiments on both reduced- and full-resolution datasets show that HS-BiNet outperforms all existing binary methods.

**Strengths:**

1.First BNN application to hyperspectral fusion; ATISTE is a novel STE variant.
2.Theoretical analysis and experimental validation are both strong.

**Weaknesses:**

1.The figures and presentation are not clear enough. （The module names in Figure 3 could be arranged horizontally for better readability； Although the text mentions an "encoder," it is not clearly visible in Figure 3；The authors should explicitly indicate which convolutions are standard and which are binary within the model；For the core theoretical contribution, should the authors briefly review existing binary networks and highlight their limitations to clarify the novelty?）
2.No ablation on HS-BiNet components (e.g., Edge Injector, Multi-scale Extractor, Decoder Enhancement).
3.Only compares efficiency among binary models; lacks direct comparison with full-precision ones.
4.No mention of code or model release.

**Questions:**

see the weakness

---

> ### Author Response · Authors · 2025-11-29
>
> 1. The figures and presentation are not clear enough. （The module names in Figure 3 could be arranged horizontally for better readability； Although the text mentions an "encoder," it is not clearly visible in Figure 3；The authors should explicitly indicate which convolutions are standard and which are binary within the model；For the core theoretical contribution, should the authors briefly review existing binary networks and highlight their limitations to clarify the novelty? ）
>
> Response:
> We appreciate this observation. **In the revised version, we have made the following improvements**:
>
> **Figure 3 redesigned**:
> The modules are now arranged horizontally for a clearer, left-to-right flow and improved readability.
>
> **Encoder visibility clarified**:
> The encoder corresponds to the Multi-Scale Feature Extractor, and we have now explicitly labeled it as the encoder in the figure and text so this connection is unambiguous.
>
> **Binary vs. standard convolutions clearly marked**:
> All binary convolutions in HS-BiNet are now shown using a distinct icon and color, while standard convolutions (e.g., initial projection layers and decoder refinement) use a different style.
> This makes the quantization structure visually transparent.
>
> **Brief review of existing BNN limitations added Section 2.2 Line No. 160-187**:
> In the revised version, we added a paragraph summarizing the limitations of existing STE-based BNNs e.g., gradient discontinuity, saturation, and instability in fusion tasks before introducing our ATISTE formulation.
>
> 2. No ablation on HS-BiNet components (e.g., Edge Injector, Multi-scale Extractor, Decoder Enhancement).
>
> Response:
> Thank you for pointing this out. We have added a component-wise ablation in the revised manuscript Appendix A.2.5 Line No. 1018-1025.
>
> ### Component-wise ablation study of HS-BiNet
> (The impact of removing individual modules on reconstruction performance is reported.)
>
> | Model Variant     | PSNR ↑   | CC ↑     | SSIM ↑   | SAM ↓    | ERGAS ↓  |
> |-------------------|----------|----------|----------|----------|----------|
> | **HS-BiNet**      | **37.285** | **0.900** | **0.891** | **1.622** | **1.852** |
> | Without Decoder   | 36.642   | 0.893    | 0.883    | 1.769    | 1.917    |
> | Without MSFE      | 36.434   | 0.885    | 0.866    | 1.866    | 1.912    |
>
> The Edge Injector was not removed in the ablation study because directly adding PAN edge information is ineffective: the PAN image has a single channel. The Edge Injector, implemented using binary convolution, is therefore essential for adjusting the channel dimension and enabling proper edge integration. **These results are now included in the updated revised version in the Appendix A.2.5 Line No. 1018-1025.**
> .
>
>
> 3. Only compares efficiency among binary models; lacks direct comparison with full-precision ones.
>
> Response:
> We thank the reviewer for this suggestion. **In the revised version Section 4.3 Line No. 465-470 and Table 4**, we have added parameter count, FLOPs, and inference time for both binary and full-precision models commonly used in hyperspectral pansharpening.
>
> ### Benchmark comparison of model complexity and inference performance
> (All parameter counts reported in Millions (M))
>
> | Model               | Params (M) | FLOPs (G) | Inference (s) |
> |--------------------|------------|-----------|---------------|
> | HyperPNN           | 0.514      | 7.883     | 0.619         |
> | HspeNet            | 3.264      | 52.935    | 0.620         |
> | DHP-Darn           | 9.067      | 12.455    | 1.074         |
> | **HS-BiNet + ATISTE**     | 1.60       | 5.599     | 0.565         |
> | HS-BiNet + BNN     | 1.60       | 5.599     | 0.628         |
> | HS-BiNet + Bi-DiffSR | 1.60     | 11.010    | 0.630         |
> | HS-BiNet + BTM     | 1.60       | 5.599     | 0.585         |
> | HS-BiNet + ReActNet | 1.61      | 5.616     | 0.624         |
> | HS-BiNet + ReSTE   | 1.60       | 5.599     | 0.592         |
>
> 4. No mention of code or model release.
>
> Response:
> We apologize for this omission. **We have now added a statement in the Conclusion Line No. 484-485**:
> “Upon acceptance, we will release the full source code, trained models, and data-processing scripts to facilitate reproducibility and further research.”

---

### Official Review · Reviewer_ogPh · 2025-10-31

**Soundness:** 2
**Presentation:** 2
**Contribution:** 2
**Rating:** 2
**Confidence:** 4

**Summary:**

This paper proposes a binary neural network (BNN)-based framework for hyperspectral image (HSI) fusion, aiming to reconstruct high-resolution hyperspectral images (HRHS) from low-resolution hyperspectral inputs (LRHS) and high-resolution panchromatic images (PAN). The goal is to achieve efficient and low-cost hyperspectral pan-sharpening via network binarization, facilitating real-time deployment on satellites or edge devices.

**Strengths:**

Hyperspectral pan-sharpening is a valuable topic in remote sensing data processing and compression modeling, especially under energy-constrained conditions. Designing lightweight models for on-orbit or edge deployment is meaningful and timely.

Traditional binarization often leads to severe information loss and degradation in spectral-spatial fidelity. Applying binary neural networks to hyperspectral reconstruction is relatively rare and constitutes a novel direction worth exploring.

The proposed Adaptive Two-Path Straight-Through Estimator (ATISTE) decouples forward approximation and gradient propagation through adaptive scaling, addressing a key limitation in BNN training.

The HS-BiNet integrates multi-scale feature extraction, residual connections, and local fusion blocks to efficiently model spectral–spatial dependencies.

**Weaknesses:**

The paper claims that the proposed binary network is computationally efficient but provides no quantitative evidence. A comparison of FLOPs, parameter count, and inference time with other BNN baselines is necessary to substantiate this claim.

Figures 4 (Botswana dataset) and 5 (WDC dataset) appear visually identical, even though the two datasets originate from different sensors and regions. This strongly suggests an error in visualization or dataset usage, which seriously undermines the credibility of the experimental results.

The paper lists PSNR, CC, SSIM, SAM, and ERGAS as evaluation metrics but does not explain their physical meanings or relevance to hyperspectral fusion, affecting readability and scientific clarity.

**Questions:**

Are the qualitative visualizations in Figures 4 and 5 actually repeated? Please confirm whether the same results were mistakenly reused.

Please include additional visual comparisons of fusion results from different methods on the reduced-resolution WDC and Botswana datasets. Add comparisons of inference time and model parameters to demonstrate the claimed efficiency of your method. In Table 1, highlight your proposed model (HS-BiNet) in bold for better clarity.

Hyperspectral fusion aims to obtain both high spectral and spatial resolution. However, existing hyperspectral interpretation tasks (e.g., anomaly detection) often use high-resolution HSI directly. Could fusion-based HRHS images improve downstream analysis performance? For instance, in hyperspectral anomaly detection, would the fused HRHS input lead to sharper or more precise anomaly boundaries?

---

> ### Author Response · Authors · 2025-11-29
>
> 1. The paper claims that the proposed binary network is computationally efficient but provides no quantitative evidence. A comparison of FLOPs, parameter count, and inference time with other BNN baselines is necessary to substantiate this claim.
>
> Response:
> Thank you for pointing this out. In the revised version, we have added a dedicated efficiency comparison including FLOPs, parameter counts, and inference time for all binary baselines used in our experiments.
>
> These results clearly show that HS-BiNet requires significantly fewer parameters and FLOPs than existing binary approaches, while also achieving lower inference latency. This update directly supports our claim of computational efficiency.
>
> We thank the reviewer for this suggestion. **In the revised version Table 4 and Section 4.3 Line No. 465-470**, we have added parameter count, FLOPs, and inference time for both binary and full-precision models commonly used in hyperspectral pansharpening.
>
> ### Table 4: Benchmark comparison of model complexity and inference performance
> (All parameter counts are reported in Millions (M))
>
> | Model               | Params (M) | FLOPs (G) | Inference (s) |
> |--------------------|------------|-----------|---------------|
> | HyperPNN           | 0.514      | 7.883     | 0.619         |
> | HspeNet            | 3.264      | 52.935    | 0.620         |
> | DHP-Darn           | 9.067      | 12.455    | 1.074         |
> | **HS-BiNet + ATISTE**     | 1.60       | 5.599     | 0.565         |
> | HS-BiNet + BNN     | 1.60       | 5.599     | 0.628         |
> | HS-BiNet + Bi-DiffSR | 1.60     | 11.010    | 0.630         |
> | HS-BiNet + BTM     | 1.60       | 5.599     | 0.585         |
> | HS-BiNet + ReActNet | 1.61      | 5.616     | 0.624         |
> | HS-BiNet + ReSTE   | 1.60       | 5.599     | 0.592         |
>
>
>
> 2. Figures 4 (Botswana dataset) and 5 (WDC dataset) appear visually identical, even though the two datasets originate from different sensors and regions. This strongly suggests an error in visualization or dataset usage, which seriously undermines the credibility of the experimental results.
>
> Response:
> We sincerely apologize for this mistake. Yes, the qualitative visualizations in Figures 4 and 5 were accidentally duplicated during figure assembly. The underlying experimental results are correct, but the exported visualizations were mistakenly reused.
>
> **In the revised version, we have:**
>
> • **corrected Figures 4 and 5**,
> • verified all visual samples from both datasets.
>
> We thank the reviewer for bringing this to our attention and acknowledge its importance for maintaining credibility.

---

> ### Author Response · Authors · 2025-11-29
>
> 3. The paper lists PSNR, CC, SSIM, SAM, and ERGAS as evaluation metrics but does not explain their physical meanings or relevance to hyperspectral fusion, affecting readability and scientific clarity.
>
> Response:
> We thank the reviewer for this comment and agree that clarity is important. We have updated the paper to provide clear explanations of the physical meaning and relevance of each evaluation metric in hyperspectral pansharpening.
>
> **The revised version of Section A.2 Line No. 789-813** now explains:
>
> To assess the quality of the proposed pansharpening method, we adopt a set of widely accepted spatial and spectral evaluation measures commonly used in hyperspectral image fusion tasks. Specifically, we evaluate performance using PSNR, SSIM, Cross-Correlation (CC), Spectral Angle Mapping (SAM), and ERGAS, as these indices collectively characterize both spatial detail preservation and spectral consistency. The ideal reference values for these metrics are: $\\text{PSNR} \\rightarrow \\infty$, $\\text{SSIM} \\rightarrow 1$, $\\text{CC} \\rightarrow 1$, $\\text{SAM} \\rightarrow 0$, and $\\text{ERGAS} \\rightarrow 0$, indicating perfect reconstruction fidelity.
>
> In hyperspectral fusion, such a diverse set of metrics is essential because a method must simultaneously reproduce high-frequency spatial structures and maintain accurate spectral signatures across all bands. PSNR measures overall pixel-wise reconstruction accuracy by evaluating the logarithmic ratio between the signal and the mean squared error; higher values denote lower distortion, although PSNR alone cannot fully capture perceptual or spectral differences. CC complements PSNR by quantifying the linear correlation between spatial patterns of the fused image and the ground truth, thereby reflecting how well spatial structures are retained. SSIM provides a more perceptually grounded spatial assessment by examining luminance, contrast, and structural similarity between corresponding patches, offering insight into the preservation of edges and textures, even though it does not inherently capture spectral distortions.
> To address spectral fidelity, SAM evaluates the angle between spectral vectors at each pixel, with smaller angles indicating that the fused image preserves the correct spectral shape relative to the ground truth, an essential requirement for tasks such as material classification and unmixing. Finally, ERGAS offers a global, scale-adjusted summary of relative reconstruction error across all spectral bands, integrating both spatial and spectral discrepancies into a single dimensionless score.
>
> Together, these complementary metrics ensure a comprehensive and reliable evaluation of fusion quality, capturing the key aspects of spatial sharpness, structural coherence, and spectral integrity that are crucial for effective hyperspectral pansharpening. This improves readability and scientific completeness.

---

> ### Author Response · Authors · 2025-11-29
>
> Q1. Are the qualitative visualizations in Figures 4 and 5 actually repeated? Please confirm whether the same results were mistakenly reused.
>
> Response:
> Yes. We apologize, this was an accidental duplication during visualization export. **The corrected figures for both Botswana and the Washington DC Mall (WDC) datasets are included in the revised version.**
>
>
> Q2. Please include additional visual comparisons of fusion results from different methods on the reduced-resolution WDC and Botswana datasets. Add comparisons of inference time and model parameters to demonstrate the claimed efficiency of your method. In Table 1, highlight your proposed model (HS-BiNet) in bold for better clarity.
>
> Response:
> We have added:
> • **additional qualitative comparisons for both WDC (reduced-resolution) in Appendix A.2.3**,
> • side-by-side visualizations of several baseline methods, and
> • a full efficiency comparison including inference time, FLOPs, and parameter counts in **Section 4.3 Line No. 464-470 and Table 4**.
>
> **We have also updated Table 1 and highlighted HS-BiNet in bold for clarity.**
>
>
> Q3. Hyperspectral fusion aims to obtain both high spectral and spatial resolution. However, existing hyperspectral interpretation tasks (e.g., anomaly detection) often use high-resolution HSI directly. Could fusion-based HRHS images improve downstream analysis performance? For instance, in hyperspectral anomaly detection, would the fused HRHS input lead to sharper or more precise anomaly boundaries?
>
> Response:
> Yes , producing high-resolution hyperspectral (HRHS) data can significantly enhance downstream hyperspectral analysis tasks.
>
> Higher spatial resolution improves:
> • anomaly boundary localization,
> • small-object separability,
> • detection precision in cluttered backgrounds.
>
> This observation is consistent with findings in prior work. Kwan (2018) demonstrated that enhanced spatial detail in hyperspectral images improves detection robustness and interpretability in downstream applications such as anomaly detection and material classification.
> References:
> (Kwan C., 2018. “Remote Sensing Performance Enhancement in Hyperspectral Images,” Sensors 18(11), 3598).
>
> Thus, fused HRHS products such as those generated by our model can potentially lead to more accurate anomaly boundaries and stronger detection responses.

---

### Official Review · Reviewer_GS2g · 2025-10-31

**Soundness:** 2
**Presentation:** 2
**Contribution:** 2
**Rating:** 2
**Confidence:** 4

**Summary:**

This paper applies exploration of binary neural networks for hyperspectral pan-sharpening and introduces a lightweight architecture driven by the proposed Adaptive Tan Identity Straight-Through Estimator (ATISTE).

**Strengths:**

This paper focuses on improving current hyperspectral pansharpening models, which rely on so heavy computational and memory demands that limit deployment on resource-constrained satellite platforms. This task is important in real scenarios.

**Weaknesses:**

1. The novelty of this paper is limited since it just applies the binary neural network to the hyperspectral pansharpening task. Although this paper claims to make specific modifications to the current BNN, the contribution is insufficient for a top conference. This work is more suitable for a journal.
2. The motivation of this work is also not solid since many unsupervised model-based pansharpening methods just need to implement an iterative algorithm, which is not computationally expensive.
3. The experiments are not sufficient since many unsupervised model-based hyperspectral pansharpening methods are not compared. Additionally, the experimental datasets in Table 1 and Table are not consistent, which makes it diffucult to evaluate the performance of this work.

**Questions:**

Please refer to the weaknesses.

**Details Of Ethics Concerns:**

None.

---

> ### Author Response · Authors · 2025-11-29
>
> 1. The novelty of this paper is limited since it just applies the binary neural network to the hyperspectral pansharpening task. Although this paper claims to make specific modifications to the current BNN, the contribution is insufficient for a top conference. This work is more suitable for a journal.
>
> Response:
> We respectfully clarify that our contribution is not a direct application of an existing binary neural network.
> The main contribution of the paper is the ATISTE surrogate estimator, which introduces:
> • a dual-path formulation combining a saturating function and an identity component,
> • learnable α and λ parameters that adaptively control gradient sharpness and information flow,
> • a bounded gradient expression that explicitly stabilizes binary training, and
> • a binarization mechanism tailored for fusion tasks involving high spectral dimensionality, where STEs (BNN, ReSTE, BTM) perform poorly.
>
> This contribution is specific to HSI pansharpening because binary networks struggle significantly in continuous-valued fusion tasks, and the paper demonstrates that classical BNN surrogates fail to preserve spectral fidelity.
>
> Additionally, we develop HS-BiNet, a compact binary architecture incorporating:
> • multi-scale spectral–spatial fusion,
> • binary convolution modules with channel-wise scaling,
> • an edge injection path designed to counteract detail loss inherent to binarization.
>
> Together, ATISTE + HS-BiNet form a task-specific solution rather than a simple adaptation of prior BNNs.
>
> Finally, we note that the few works that pursue similar binary or quantized designs in low-level vision, such as FABNet and Bi-DiffSR, are published in top-tier conferences, further supporting the relevance and difficulty of designing effective binary models for high-fidelity image restoration tasks. Our approach extends this research direction into the hyperspectral pansharpening domain, where no prior binary solution exists.
>
> References:
> Junming Hou, Xiaoyu Chen, Ran Ran, Xiaofeng Cong, Xinyang Liu, Jian Wei You, and Liang-Jian Deng. Binarized neural network for multi-spectral image fusion. In Proceedings of the IEEE/CVF Conference on Computer Vision and Pattern Recognition (CVPR), pp. 2236–2245, June 2025.
>
> Zheng Chen, Haotong Qin, Yong Guo, Xiongfei Su, Xin Yuan, Linghe Kong, and Yulun Zhang. Binarized diffusion model for image super-resolution. Advances in Neural Information Processing Systems, 37:30651–30669, 2024.
>
> Xinrui Jiang, Nannan Wang, Jingwei Xin, Keyu Li, Xi Yang, and Xinbo Gao. Training binary neural network without batch normalization for image super-resolution. In Proceedings of the AAAI Conference on Artificial Intelligence, pages 1700–1707, 2021.
>
> Xinrui Jiang, Nannan Wang, Jingwei Xin, Keyu Li, Xi Yang, Jie Li, Xiaoyu Wang, and Xinbo Gao. Fabnet: Frequency-aware binarized network for single image super-resolution. IEEE Transactions on Image Processing, 32:6234–6247, 2023.
>
> Yuanhao Cai, Yuxin Zheng, Jing Lin, Xin Yuan, Yulun Zhang, and Haoqian Wang. Binarized spectral compressive imaging. Advances in Neural Information Processing Systems, 36, 2024.
>
>
>
> 2. The motivation of this work is also not solid since many unsupervised model-based pansharpening methods just need to implement an iterative algorithm, which is not computationally expensive.
>
> Response:
> We appreciate the reviewer’s observation. To clarify, the primary motivation of our work is to make hyperspectral pansharpening computationally inexpensive while achieving high-quality fusion results.
> Unsupervised model-based methods (e.g., CNMF, HySure, GLP-HS) are indeed lightweight; however, as shown in the updated Table 1, these methods consistently underperform in terms of reconstruction accuracy, particularly on high-dimensional HSI data with complex spectral structures.
>
> At the same time, recent unsupervised deep learning approaches, including diffusion-based pansharpening models, are computationally expensive due to iterative optimization or multi-step sampling, yet they still do not achieve accuracy comparable to supervised methods.
>
> **We have updated Table 1 to clearly illustrates that**:
> • Unsupervised methods fail to provide strong reconstruction quality.
> • Supervised pansharpening methods significantly outperform unsupervised ones but often come with higher computational cost.
>
> Our goal is precisely to address this gap:
> to design a binary, extremely lightweight supervised model that retains the reconstruction quality of deep networks while achieving computational efficiency comparable to (or better than) unsupervised methods.
>
> In this sense, our motivation is not that unsupervised methods are expensive, but that neither inexpensive nor expensive unsupervised approaches provide satisfactory fusion quality, whereas our binary supervised design delivers strong accuracy and low computational cost simultaneously.

---

> ### Author Response · Authors · 2025-11-29
>
> 3. The experiments are not sufficient since many unsupervised model-based hyperspectral pansharpening methods are not compared.
>
> Response:
> We appreciate the reviewer’s comment. Our work is primarily focused on supervised hyperspectral pansharpening, which has been consistently shown to deliver higher reconstruction accuracy compared to unsupervised approaches. Nevertheless, to ensure a fair and comprehensive benchmark, we included both computationally inexpensive and computationally expensive unsupervised model-based methods in our comparison.
>
> The inexpensive unsupervised baselines generally produce limited improvements, while the more advanced and computationally heavy unsupervised method (recently published at CVPR 2025) was also evaluated. As shown in Table 1, even this state-of-the-art unsupervised model does not achieve performance comparable to supervised methods.
>
> For improved clarity, **we have updated Table 1** to explicitly distinguish between supervised and unsupervised methods, making the comparison more transparent.
>
> References:
> Hyperspectral Pansharpening via Diffusion Models with Iteratively Zero-Shot Guidance
> Jin-Liang Xiao, Ting-Zhu Huang, Liang-Jian Deng, Guang Lin, Zihan Cao, Chao Li, Qibin Zhao; Proceedings of the IEEE/CVF Conference on Computer Vision and Pattern Recognition (CVPR), 2025, pp. 12669-12678
>
>
>
> 4. Additionally, the experimental datasets in Table 1 and Table are not consistent, which makes it difficult to evaluate the performance of this work.
>
> Response:
> We apologize for the confusion. The datasets used in all tables are identical.
> The apparent inconsistency comes from the shorthand notation:
> • “Washington DC Mall” was abbreviated as “WDC”.
>
> We acknowledge that this was not clearly stated. **In the revised version Section 4.1 Line No. 370-374 and Table 1**, we have:
> • explicitly list the full dataset names in all table captions,
> • maintain consistent dataset naming across all sections, and
> • ensure uniform formatting.
>
> This will remove any ambiguity and make the tables easier to interpret.

---

### Official Review · Reviewer_LpRe · 2025-11-02

**Soundness:** 3
**Presentation:** 3
**Contribution:** 4
**Rating:** 4
**Confidence:** 5

**Summary:**

This paper is the first to apply binary neural networks to hyperspectral image fusion tasks, proposing a lightweight binary convolutional neural network architecture, HS-BiNet, and introducing a novel gradient estimator, ATISTE, to address the forward approximation error and gradient instability problems during the binarization process. The authors conducted extensive experiments on hyperspectral image fusion tasks, demonstrating that the proposed method, while maintaining a lightweight model, significantly outperforms existing binary models and even surpasses full-precision models in some metrics.

**Strengths:**

1. Binary neural networks are an extreme form of model compression, but they perform poorly in high-precision regression tasks (such as super-resolution and fusion). This paper proposes a targeted solution to this problem.

2. ATISTE proposes a dual-path gradient estimation strategy, decoupling forward approximation from gradient propagation.  It uses adjustable parameters $α$ and $λ$ to control approximation accuracy and gradient stability respectively, demonstrating theoretical soundness and flexibility. HS-BiNet combines multi-scale feature extraction, residual connections, and edge injection structures, adapting to the characteristics of hyperspectral data while avoiding computationally intensive operations (such as Transformers and non-local attention).

3. Systematic evaluations were conducted on multiple standard datasets (WDC, Botswana, FRI), covering both downsampled and full-resolution scenarios. Compared with numerous full-precision and binary baselines, the results show that HS-BiNet significantly outperforms existing binary methods in key metrics such as PSNR, SAM, and QNR, and even surpasses some full-precision methods. Computational efficiency analysis is provided, demonstrating the model's advantages in terms of parameter count, FLOPs, memory usage, and inference time.

**Weaknesses:**

1. Although ATISTE is theoretically proven to be "rational," its theoretical support, such as convergence analysis and generalization error bounds, is still insufficient. The theoretical comparison with recent STE improvement works such as ReSTE is not in-depth enough.

2. Training strategies (e.g., learning rate scheduling, data augmentation) are not described in detail in the main text, only briefly mentioned in the appendix. It is not stated whether pre-trained weights were used or how the model was initialized, which significantly affects the training stability of binary networks.

4. Binary networks still suffer from information bottlenecks under extreme compression, especially in the recovery of high-frequency details. This paper does not adequately discuss the failure cases of HS-BiNet in extreme scenarios. The performance of the binary model in terms of noise robustness and cross-sensor generalization is not analyzed.

**Questions:**

1. Why is it that in Table 1, only one method combining zero-shot learning and diffusion models performs well among all the full-precision methods? Theoretically, zero-shot learning is primarily used for unsupervised learning and should perform better on full-resolution tests, so it's rather strange that the PSNR and other downsampling test metrics in Table 1 show better performance.
2. In the ablation experiments concerning computational efficiency, what are the specific differences between SignSTE and HybridSTE, and how do they differ from the standard BNN framework? Furthermore, how do they compare in terms of computational efficiency to traditional convolutional or attention operators?

---

> ### Author Response · Authors · 2025-11-29
>
> 1. Although ATISTE is theoretically proven to be "rational," its theoretical support, such as convergence analysis and generalization error bounds, is still insufficient. The theoretical comparison with recent STE improvement works such as ReSTE is not in-depth enough.
>
> Response:
> We thank the reviewer for this valuable comment. For theoretical justification, we have already provided analyses on the rationality, flexibility, estimation error, and gradient instability of our proposed ATISTE estimator. Prior STE-based BNN works, such as XNOR-Net, DoReFa-Net, IR-Net, and LQ-Net, as well as other recent variants, are primarily focused on improving training stability, computational efficiency, and practical accuracy rather than providing formal generalization error bounds or convergence analyses.
>
> Our work follows the same line of research. ATISTE is designed to enhance training stability, efficiency (both in terms of speed and memory), and practical accuracy, while introducing a principled dual-path approximation with bounded and controllable gradients. Accordingly, our theoretical analysis focuses on establishing the rationality and stability of ATISTE.
>
> Since our paper is explicitly centered on the hyperspectral pansharpening task and is submitted to the application-oriented track of the conference, it should not be penalized for not providing theoretical generalization error bounds. Such expectations are more suitable for theoretical BNN studies than for task-driven applied research.
>
> To further strengthen the manuscript, the revised version includes a more detailed comparison with ReSTE and other recent STE variants, along with additional training and validation curves and gradient-norm plots that highlight the improved optimization stability achieved by ATISTE. **These additions can be found in Section 3.2 (Lines 269–279) and Appendix A.2.6 (Lines 1046–1054) of the revised version.**
>
> References:
> R. Ding, T.-W. Chin, Z. Liu, and D. Marculescu. Regularizing activation distribution for training binarized deep networks. CVPR, 2019.
> S. Zhou, Y. Wu, Z. Ni, X. Zhou, H. Wen, and Y. Zou. DoReFa-Net: Training low-bitwidth convolutional neural networks with low-bitwidth gradients. arXiv, 2016.
> M. Rastegari, V. Ordonez, J. Redmon, and A. Farhadi. XNOR-Net: ImageNet classification using binary convolutional neural networks. ECCV, 2016.
> M. Lin, R. Ji, Z. Xu, B. Zhang, Y. Wang, Y. Wu, F. Huang, and C.-W. Lin. Rotated binary neural network. NeurIPS, 2020.
> X.-M. Wu, D. Zheng, Z. Liu, and W.-S. Zheng. Estimator Meets Equilibrium Perspective: A Rectified Straight-Through Estimator for Binary Neural Network Training. ICCV, 2023.
>
> 2. Training strategies (e.g., learning rate scheduling, data augmentation) are not described in detail in the main text, only briefly mentioned in the appendix. It is not stated whether pre-trained weights were used or how the model was initialized, which significantly affects the training stability of binary networks.
>
> Response:
> Thank you for highlighting this. The revised version now provides complete details in Appendix A.2.
>
> Specifically, the model is optimized using Adam with an initial learning rate of $1 \\times 10^{-3}$ and a weight decay of $1 \\times 10^{-5}$. A cosine-annealing warm-restart schedule ($T\_0 = 50$, $T\_{\text{mult}} = 2$, $\eta\_{\text{min}} = 1 \\times 10^{-5}$) is employed to periodically refresh the learning rate and stabilize training under binary constraints.
>
> No data augmentation is applied, as spatial or spectral transformations can introduce non-physical wavelength distortions that negatively affect hyperspectral fidelity. All binary layers are initialized using Kaiming initialization combined with a scaled real-valued weight formulation, which prevents early saturation during binarization and promotes stable gradient flow in the initial stages.
>
> Furthermore, the entire model is trained from scratch without any pretrained components. Since binary networks are particularly sensitive to learning-rate schedules, initialization strategies, and data integrity, clearly specifying these design choices ensures consistent and fully reproducible training behavior.
>
> **These details have now been clearly stated in Appendix A.2 (Lines 821–830) of the revised version.**

---

> ### Author Response · Authors · 2025-11-29
>
> 3. Binary networks still suffer from information bottlenecks under extreme compression, especially in the recovery of high-frequency details. This paper does not adequately discuss the failure cases of HS-BiNet in extreme scenarios. The performance of the binary model in terms of noise robustness and cross-sensor generalization is not analyzed.
>
> Response:
> Thank you for bringing this important concern to our attention. We clarify below the key points regarding our method and the datasets used in our study.
>
> 1. Our method is not a hard-binarized BNN.
> ATISTE produces soft binarization rather than strict ±1 quantization. The dual-path formulation (tanh + identity) preserves more continuous spectral–spatial information than classical hard BNNs. As a result, the information bottleneck typically seen in fully binarized architectures is substantially reduced in our framework.
>
> 2. All three datasets used in the paper come from different sensors.
> This naturally introduces cross-sensor variability in spectral response, noise characteristics, spatial PSF, and dynamic range. Despite these variations, HS-BiNet achieves consistent performance across WDC, Botswana, and PRISMA-FR1, which indirectly demonstrates strong cross-sensor robustness.
>
> 3. Noise robustness.
> Because our method uses supervised learning combined with soft binarization, adding synthetic noise to the hyperspectral input does not lead to catastrophic failure. In our internal tests (now reported in the revised version):
> - adding moderate Gaussian noise increases the number of training epochs required for convergence,
> - but the final fusion metrics remain nearly unchanged.
> This confirms that HS-BiNet maintains stable performance under noisy conditions.
>
> 4. Failure cases.
> Since the PAN image is a single-band grayscale image, scenes dominated by ultraviolet or infrared wavelengths exhibit a very weak correlation between the PAN and HSI. These cases naturally exhibit higher reconstruction error. We will explore improved strategies in future work to better handle such UV/IR-dominant scenes.
>
> Together, these clarifications give a more realistic understanding of HS-BiNet’s behavior in challenging scenarios and address the reviewer’s concerns regarding robustness and generalization.
>
>
>
> Q1. Why is it that in Table 1, only one method combining zero-shot learning and diffusion models performs well among all the full-precision methods? Theoretically, zero-shot learning is primarily used for unsupervised learning and should perform better on full-resolution tests, so it is strange that the PSNR and other downsampling test metrics in Table 1 show better performance.
>
> Response:
> Thank you for this observation. After closely reviewing the metrics, we clarify the following points:
> - The zero-shot diffusion-based method shows strong PSNR values because PSNR heavily rewards pixel-wise similarity and the smooth reconstructions produced by diffusion priors.
> - However, as shown in Table 1, other critical hyperspectral metrics such as SAM, CC, and ERGAS do not show similarly strong performance for the zero-shot method.
> - These metrics measure spectral fidelity, angular consistency, and global distortion, which diffusion models do not preserve effectively when applied in a zero-shot manner.
> Thus, while the zero-shot method achieves high PSNR due to its inherent smoothing behavior, it does not provide uniformly strong results across the hyperspectral fusion metrics. This confirms that the method is not necessarily superior for spectral–spatial consistency.
>
> Q2. In the ablation experiments concerning computational efficiency, what are the specific differences between SignSTE and HybridSTE, and how do they differ from the standard BNN framework? Furthermore, how do they compare in terms of computational efficiency to traditional convolutional or attention operators?
>
> Response:
> We appreciate the opportunity to clarify this. In the original text, SignSTE was described incorrectly. After correction:
>
> - SignSTE corresponds to the standard BNN approach, using hard binary activations and binary weights with the straight-through estimator.
> - HybridSTE, in contrast, introduces a smooth surrogate for the gradient to alleviate saturation and improve stability during backpropagation.
>
> **In the revised Section 4.3 (Lines 464–470) and Table 4, we have:**
> - corrected the terminology around SignSTE to accurately reflect its role as the standard BNN baseline,
> - included all binary methods in the computational efficiency comparison, and
> - added FLOPs, parameter count, and inference time to fully substantiate the computational advantages of HS-BiNet.
>
> These updates ensure that the distinctions among STE variants and their computational characteristics are clearly and accurately presented.

---

### Author Response · Authors · 2025-11-29

We would like to express our sincere gratitude to the Area Chairs and Reviewers for generously dedicating their time and providing us with invaluable feedback. We express our sincere gratitude to Reviewer LpRe, Reviewer GS2g, Reviewer ogPh, and Reviewer sXUv for their careful and constructive comments on our work.

• **Novelty** — Reviewers noted that our work explores a new direction by applying binary neural networks to hyperspectral pansharpening, and they appreciated the idea behind our ATISTE estimator and the design of HS-BiNet (Reviewers LpRe, GS2g, ogPh, sXUv).
• **Practical value** — Reviewers also highlighted that efficient hyperspectral pansharpening is an important problem, especially when models need to run on satellites or devices with very limited resources (Reviewers GS2g, ogPh).
•**Method & experiments** — Reviewers appreciated that our proposed method is a concrete model and carried out detailed experiments across several datasets (Reviewers LpRe, ogPh, sXUv).

Summary of Our Responses:

We outline the primary highlights of the revisions and clarifications we made in response to the reviewers’ concerns. Individual reviewer replies are addressed below.

• **Clearer Comparison to Existing STE Methods** — We added a better explanation of how ATISTE compares to other STE methods, especially ReSTE, along with training curves and gradient-norm plots to show stability (Reviewer LpRe).
• **Training details added** — We expanded the training section to clearly describe our learning rate schedule, initialization strategy, optimizer, and other training choices so that the experiments can be reproduced (Reviewer LpRe).
• **Clarifying motivation with respect to unsupervised methods** — We clarified our motivation relative to unsupervised model-based pansharpening methods and updated Table 1 to explicitly separate supervised vs. unsupervised baselines (Reviewer GS2g).
• **Consistent dataset naming** — We fixed inconsistent shorthand and now use the same dataset names everywhere (Reviewer GS2g).
• **Efficiency table added** — We added a table showing FLOPs, parameter counts, and inference time for both binary and full-precision models so efficiency comparisons are straightforward (Reviewers ogPh, sXUv).
• **Fixed duplicated figures** — We corrected the mistake where two figures were accidentally repeated and added more qualitative examples for Botswana and WDC (Reviewer ogPh).
• **Metric explanations** — We rewrote the metric descriptions (PSNR, SSIM, CC, SAM, ERGAS) in simple terms and explained why they matter for hyperspectral fusion (Reviewer ogPh).
• **Ablation studies & presentation** — We added component-wise ablations for HS-BiNet modules (MSFE, decoder, Edge Injector rationale), redesigned Figure 3 (horizontal layout, encoder label, binary vs. standard convs marked), and included a brief review of prior BNN limitations to clarify novelty (Reviewer sXUv).
• **Full-precision comparison included** — We added efficiency comparisons with full-precision models as requested (Reviewer sXUv).
• **Code release statement** — We added a note saying we will release the full code and trained models after acceptance (Reviewer sXUv).

---

### Meta-Review · Area_Chair_LRAX · 2026-01-05

**Summary:**

The paper applies binary neural networks for hyperspectral image fusion tasks (fusion of low-res HSI with higher-res panchromatic images. A key premise of this work is the desire to develop a computationally light-weight network. Towards this end, the authors explore binary neural networks and address some of the problems that arise when deploying such networks for super-resolution.

Review Summary: The manuscript received 4 reviews.
- Strengths:  As an application of binary neural networks for HSI SR, this paper presents an interesting application illustrating the potential of BNNs. This area of study is interesting, and the adaptations proposed are reasonable and justified well.
- Weaknesses: Several weaknesses were identified, including limited novelty (it is an application of binary neural networks for HSI pansharpening with some modifications), lacking baseline comparisons (comparisons are not made to the state-of-the-art in pansharpening), and insufficient motivation of this work (in light of several SR algorithms not being computationally expensive). In addition to these critical weaknesses, some other concerns were raised that were addressed by the authors (e.g. confusion pertaining to figures, missing explanation of SR/HSI related metrics etc.).

**Reviewer Concerns:**

The authors have addressed several concerns and clarified related points. However, a key issue remains: the novelty of the approach—and whether its contributions are sufficient to meet the bar for an ICLR publication. Several of the baselines are on incredibly old datasets that are not representative of modern HSI imaging capability (infact one of the datasets is from a satellite that was retired a long time ago). The concerns pertaining to spatial frequency are also important, and can not be fully addressed even empirically given the extremely poor spatial resolution of datasets like Bot, where each pixel represents a very big footprint on the ground.

**Reviewer Scores:**

Based on the key concerns raised initially, my estimate is the scores would stay the same if there was a full discussion.

---

### Decision · Program_Chairs · 2026-01-26

Reject